# SPARSE AUTOENCODERS REVEAL SELECTIVE REMAPPING OF VISUAL CONCEPTS DURING ADAPTATION

**Hyesu Lim**[1,2*]   **Jinho Choi**[2]   **Jaegul Choo**[2]   **Steffen Schneider**[1,3†]

[1]Institute of Computational Biology, Computational Health Center, Helmholtz Munich
[2]KAIST AI   [3]Munich Center for Machine Learning (MCML)

## ABSTRACT

Adapting foundation models for specific purposes has become a standard approach to build machine learning systems for downstream applications. Yet, it is an open question which mechanisms take place during adaptation. Here we develop a new Sparse Autoencoder (SAE) for the CLIP vision transformer, named PatchSAE, to extract interpretable concepts at granular levels (*e.g.*, shape, color, or semantics of an object) and their patch-wise spatial attributions. We explore how these concepts influence the model output in downstream image classification tasks and investigate how recent state-of-the-art prompt-based adaptation techniques change the association of model inputs to these concepts. While activations of concepts slightly change between adapted and non-adapted models, we find that the majority of gains on common adaptation tasks can be explained with the existing concepts already present in the non-adapted foundation model. This work provides a concrete framework to train and use SAEs for Vision Transformers and provides insights into explaining adaptation mechanisms.

Code and Demo: `github.com/dynamical-inference/patchsae`

# 1   INTRODUCTION

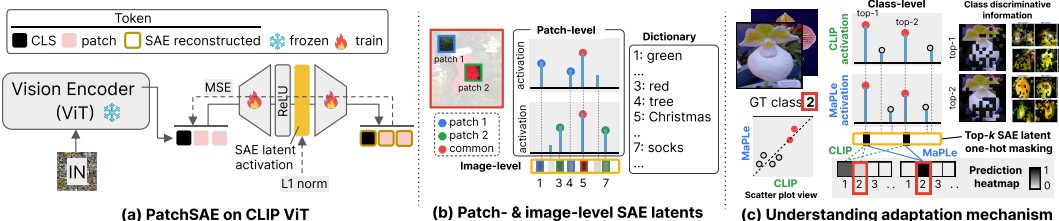

**Figure 1: Overview**. (a) We train our PatchSAE on a frozen CLIP ViT with an MSE loss and an L1 sparsity regularizer using ImageNet (IN) (§3.1). (b) We analyze the trained PatchSAE by interpreting patch- and image-level concepts of activated SAE latents (§3.2 & 3.3). (c) We then investigate the influence of SAE latents on the model behavior in classification tasks (§4.1) and explain how adaptation methods improve the downstream task performance (§4.2).

Foundation models excel at fast adaptation to new tasks and domains with limited extra training data (Radford et al., 2021; Caron et al., 2021; Touvron et al., 2023). In the space of vision-language models, CLIP (Radford et al., 2021) became an important backbone for numerous applications (Liu et al., 2024; Li et al., 2023; Rombach et al., 2022). The CLIP model consists of two transformer networks to encode text and image inputs. Various parameter-efficient adaptation techniques, such as adopting learnable tokens, have been proposed targeting either of the systems. While early works targeted the text encoder (Zhou et al., 2022b,a), more recently it was shown that joint adaptation of both the text and image encoders can further improve classification performance (Khattak et al., 2023a,b). Despite these advances in adaptation methods, it remains an open question of *how* representations in a foundation model change during adaptation.

---

*This work was done during a research visit at Helmholtz Munich.
†Correspondence: steffen.schneider@helmholtz-munich.de

Recently, Sparse Autoencoders (SAEs; Bricken et al., 2023; Cunningham et al., 2023) have gained emerging attention as a tool of mechanistic interpretability (Templeton, 2024; Yun et al., 2021) after showing its effectiveness in widely used LLMs (Bricken et al., 2023; Lieberum et al., 2024). SAEs map dense model representations, which are difficult to interpret because multiple unrelated concepts are entangled (*polysemantic*), to sparse and interpretable (*monosemantic*) concepts.

In this work, we develop PatchSAE, a new SAE model for the CLIP Vision Transformer (ViT) (Fig. 1(a)). PatchSAE extracts interpretable concepts and their patch-wise spatial attributions, providing localized understanding of multiple visual concepts that can be simultaneously captured from different regions of a single image (Fig. 1(b)). Before analyzing model behaviors through the SAE, we first validate PatchSAE as an interpretability tool. Our PatchSAE identifies diverse interpretable concepts, provides localized interpretation, and performs well across multiple datasets (§3.3). We then explore the influence of interpretable concepts on the final output of the CLIP model through classification tasks.

We use our PatchSAE to shed light on the internal mechanisms of foundation models during adaptation tasks. Recent state-of-the-art adaptation methods for CLIP (Zhou et al., 2022a,b; Khattak et al., 2023a,b) add trainable, dataset specific tokens for adaptation, akin to a system prompt in LLMs. Through extensive analysis, we reveal a wide range of interpretable concepts of CLIP, including simple visual patterns to high-level semantics, employing our PatchSAE as an interpretability tool (Fig. 1(b)). We also localize the recognized concepts through token-wise inspections of SAE latent activations, while extending it to image-, class-, and task-wise understandings. Furthermore, we demonstrate that the SAE latents have a crucial impact on the model prediction in classification tasks through ablation studies. Lastly, we show evidence that prompt-based adaptation gains the performance improvement by tailoring the mapping between recognized concepts and the learned task classes (Fig. 1(c)).

Our paper proceeds as follows: First, we introduce PatchSAE, a sparse autoencoder which allows to discover concepts for each token within a vision-transformer with spatial attributions (§3). We train this model on CLIP, and show the interpretability and generalizability of ImageNet-scale trained SAE across domain-shifted and finer-grained benchmark datasets. We explore the CLIP behavior in classification tasks and how adding learnable prompts changes the model behavior using PatchSAE (§4). In the end, we discuss conclusions and broader implications (§5).

## 2 RELATED WORK

**Sparse autoencoders for mechanistic interpretability.** Mechanistic interpretability (Elhage et al., 2021) aims to interpret how neural networks infer their outputs. To achieve this, it is natural to seek a deeper understanding of which feature (concept) is recognized by each neuron in the neural network (Olah et al., 2020). For instance, the logit lens (Nostalgebraist, 2020) approach attempts to understand the intermediate layer output by mapping it into the final classification or decoding layer. However, understanding neurons in human interpretable form is challenging due to the polysemantic (Elhage et al., 2022) nature of neurons, where each neuron activates for multiple unrelated concepts. This property is attributed to superposition (Elhage et al., 2022), where neural networks represent more features than the number of dimensions.

To overcome the superposition phenomenon in neural network interpretation, sparse autoencoders (SAEs) (Sharkey et al., 2022; Bricken et al., 2023) have recently gained significant attention. SAEs decompose model activations into a sparse latent space, representing them as dictionary of high dimensional vectors. Several studies (Yun et al., 2021; Cunningham et al., 2023) have applied SAEs to language models. Using SAEs, Templeton (2024) discovered bias- and safety-related features in large language models (LLMs), demonstrating that these features can be steered to alter the behavior of LLMs. Recent research extended the application of SAEs to vision-language models, such as CLIP (Radford et al., 2021). Fry (2024) and Daujotas (2024a) extracted interpretable concepts from the vision encoder of CLIP and Daujotas (2024b) utilized these features to edit image generation in a diffusion model. Rao et al. (2024) named the SAE concepts using word embeddings from the CLIP text encoder and used them for a concept bottleneck model.

Distinct from previous works, we propose to use patch-level image tokens for SAEs which allows intuitive and localized understanding of SAE latents and easily transformable to higher (image- / class- / dataset-) level of analysis. Furthermore, we adopt SAE latents masking method to examine

the relationship between interpretable concepts and downstream task-solving ability. For the first time, this allows precise investigation of how foundation models behave during adaptation, and how concepts are re-used across datasets.

**Prompt-based adaptation.** Adapting vision-language foundation models like CLIP through fine-tuning requires large datasets and significant computation resources. In addition, the generalization ability of the model may be compromised after fine-tuning. As an alternative, prompt-based adaptation has recently emerged, training only a few learnable tokens while keeping the weight of pre-trained models fixed. CoOp (Zhou et al., 2022b) proposed adapting CLIP in few-shot learning setting by optimizing learnable tokens in the language branch, while Bahng et al. (2022) applied prompt adaptation to the vision branch. MaPLe (Khattak et al., 2023a) improved few-shot adaptation performance by jointly adding learnable tokens to both the vision and language branches and considered as a base structure for more recent multimodal prompt adaptation methods (Khattak et al., 2023b). Although these studies demonstrate that prompt learning is effective for adapting CLIP, there is still a lack of research focusing on how and why prompt learning enables such adaptation.

Our work focuses on exploring the CLIP image encoder using an SAE on a vision transformer. We choose MaPLe (Khattak et al., 2023a) as a representative structure of multimodal prompt adaptation and investigate the internal work of adaptation methods.

## 3 PATCHSAE: SPATIAL ATTRIBUTION OF CONCEPTS IN VLMS

In this section, we revisit the basic concept of sparse SAE and introduce our new PatchSAE model in §3.1. We then discuss how we discover interpretable SAE latent directions in §3.2. Our analysis includes computing summary statistics of SAE latents, that indicate how often and how strongly one latent is activated, detecting model-recognized concepts by inspecting reference images with high SAE latent activations, and spatially localizing SAE concepts in the image space (Fig. 2).

### 3.1 PATCHSAE ARCHITECTURE AND TRAINING OBJECTIVES

SAEs typically consist of a single linear layer encoder, followed by a ReLU non-linear activation function, and a single linear layer decoder (Bricken et al., 2023; Cunningham et al., 2023). To train an SAE on CLIP Vision Transformer (ViT), we hook intermediate layer outputs from the pre-trained CLIP ViT and use them as self-supervised training data. We leverage all tokens including class (CLS) and image tokens from the residual stream output [1] (Fig. 9(c)) of an attention block and feed them to the SAE. Formally, we take ViT hook layer output as an SAE input $\mathbf{z}$, multiply it with the encoder layer weight $W_E \in \mathbb{R}^{d_{\text{ViT}} \times d_{\text{SAE}}}$, pass to the ReLU activation $\phi$, then multiply with the decoder layer weight $W_D \in \mathbb{R}^{d_{\text{SAE}} \times d_{\text{ViT}}}$ [2]. The column (or row) vectors of the encoder (or decoder), $d_{\text{SAE}}$ vectors size of $\mathbb{R}^{d_{\text{ViT}}}$, correspond to the candidate concepts, *i.e.*, *SAE latent directions*. We call the output vector of the activation layer (size of $\mathbb{R}^{d_{\text{SAE}}}$) as *SAE latent activations*. For simplicity, we use $f$ for the encoder and $g$ for the decoder:

$$\mathbf{z} = \text{ViT}(\mathbf{x})[\text{hook layer}], \qquad \text{SAE}(\mathbf{z}) = (g \circ \phi \circ f)(\mathbf{z}) = W_D^\top \phi(W_E^\top \mathbf{z}). \qquad (1)$$

To train the SAE, we minimize the mean squared error (MSE) as a reconstruction objective, and use L1 regularization on the SAE latent activations to learn sparse concept vectors (Fig. 1(a)):

$$\mathcal{L}_{\text{SAE}} = \|\text{SAE}(\mathbf{z}) - \mathbf{z}\|_2^2 + \lambda_{l_1} \|\phi(f(\mathbf{z}))\|_1. \qquad (2)$$

An ideal SAE encoder maps the dense model representations into multiple monosemantic concepts and an ideal decoder reconstructs the original vector by linearly combining these distinct concepts.

**Training details.** We use a CLIP model with an image encoder of ViT-B/16, which results in $14 \times 14$ image tokens and a CLS token as input. It has 12 attention layers with model dimension $d_{\text{ViT}}$ of 768. For PatchSAE, we set the expansion factor to 64 that multiplies with $d_{\text{ViT}}$, which results in a SAE latent dimension $d_{\text{SAE}}$ of 49,152. We take the ViT output from the residual stream of the second last attention layer (i.e., 11-th layer output). Note that we use all image tokens, so the input and output of SAE have a size of (number of samples, token length, model dimension $d_{\text{ViT}}$). We average the training loss across all individual tokens. We evaluate the trained SAE for reconstruction ability and the sparsity. We report the training performance of different configurations (§A.1) and show variations for training the SAE on different layers (§A.2) in the Appendix.

---

[1] The residual stream is the sum of the attention block's output and its input, see Fig. 9(c).

[2] We use bias terms for linear layers and centralize $\mathbf{z}$, *i.e.*, $\text{SAE}(\mathbf{z} - \mathbf{b}_{\text{dec}})$.

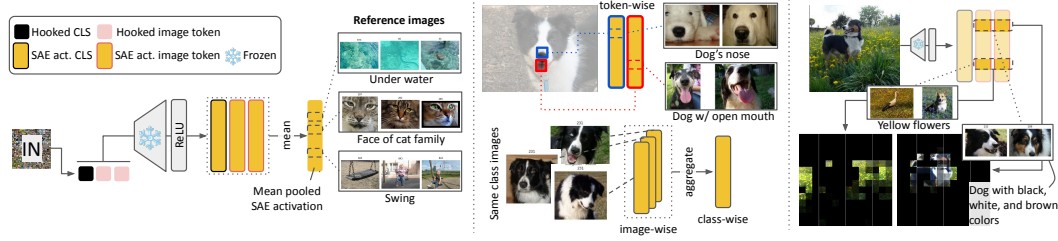

**Figure 2: Analyzing SAE latents**. (a) We take an average over patch-level activations for an image and keep top-$k$ images having the highest mean activation as the reference images for each SAE latent. (b) From patch-level latent activations, we investigate localized concepts. Furthermore, we represent image-, class-, and dataset-wise concepts by aggregating the patch-level activations. (c) For a certain concept, we provide the spatial attribution of the concept by visualizing the patch-level activations as a segmentation mask.

## 3.2 ANALYSIS METHOD AND EVALUATION SETUP

After training, we validate our PatchSAE model by interpreting the activated SAE latents from input examples. We first discover the candidate concepts – the *SAE latent directions* – by collecting reference images that maximally activate each SAE latent and compute their summary statistics. Then, we investigate the *SAE latent activation* to see how much the given input aligns with the corresponding SAE latent directions. The token-wise (*i.e.*, patch-wise) investigation allows spatially localized understandings of an image. To derive a global interpretation of the image for an SAE latent, we aggregate the patch-level activations into an image-level activation by counting the number of patches activating the corresponding latent. Similarly, we extend it to class- and dataset-level analysis (Eq. 5).

**Reference images for SAE latents.** As a first step of discovering the interpretable concepts that SAE represents, we consider a set of images that maximally activate each SAE latent as *reference images*. Given a trained SAE and a dataset, we keep the top-$k$ images having the highest SAE latent activation value for each latent dimension, *i.e.*, we have $k \times d_{\text{SAE}}$ reference images in total. Here, we use image-level activations to select top-$k$ images. Fig. 2(a) illustrates the procedure.

**Summary statistics of SAE latents.** To inform the general trend of an SAE latent, we compute summary statistics of the activation distribution (Bricken et al., 2023). We use the *activated frequency* and the *mean activation values* over a subset of training images. Using the class label information from a classification benchmark dataset such as ImageNet, we compute the *label entropy* (Fry, 2024) and *standard deviation* from the reference images. Specifically, we compute and interpret the statistics as follows:

- **Sparsity (activated frequency)** represents how frequently this latent is activated. We count the number of images having positive SAE latent activations and divide by the total number of seen images. An SAE latent with a high frequency either represents a common concept or is an uninterpretable (noisy) latent.

- **Mean activation value** is computed by averaging the positive activation value among the activated samples. The mean activation value implies the SAE model's confidence. A latent direction is more likely to represent a meaningful concept if it has a high mean activation value.

- **Label entropy** measures how many unique labels activate the latent. Precisely, we compute the probability of a label based on its activation value and compute the entropy as

$$\text{prob}_c = \frac{\text{sum}_c}{\sum_{c \in C} \text{sum}_c}, \quad \text{entropy} = -\sum_{c \in C} (\text{prob}_c \log \text{prob}_c), \quad (3)$$

  where $\text{sum}_c$ is the summed activation values for label $c \in C$. The entropy being equal to zero indicates that all reference images have exactly the same label. Higher entropy indicates that more labels contribute to the latent's activation.

- **Label standard deviation**. In ImageNet, class labels are organized in a hierarchical structure based on WordNet's semantic relationships (Deng et al., 2009; Miller, 1995). We leverage this label structure and use the label standard deviation of reference images as a clue for the semantic granularity besides the label entropy when exploring the latents. We discuss more in §A.1.

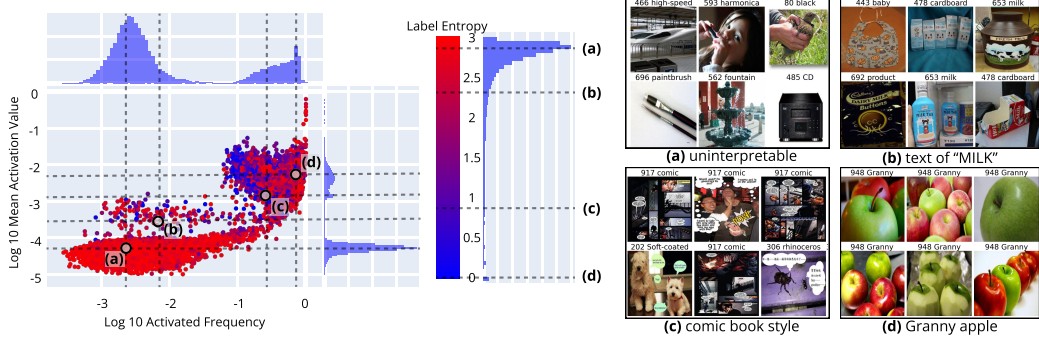

**Figure 3: SAE latents statistics and reference images.** Left: Scatter plot of SAE latent statistics ($x$-axis: log10 of activated frequency, $y$-axis: log10 of mean activation) colored by label entropy. Right: Reference images from Imagenet of four SAE latents in different regions.

**Discovering active SAE latents in diverse levels.** Using the reference images as an interpretable proxy for SAE latent directions and the SAE latent activation of an input as the similarity with the corresponding latent, we examine *which* concepts are *how strongly* active for the input. As depicted in Fig. 2(b) and (c), the *patch-level* SAE latent activations inform the recognized concepts from each patch. For example, for the input patch containing the "dog's nose", the SAE latent having the reference images of "dog's nose" is active.

To obtain a global concept from the image, we transform the patch-level activations into an *image-level* activation. In specific terms, we first binarize the SAE latent activation of the $i$-th image at $j$-th token for the $s$-th SAE latent $\mathbf{h}_{i,j}[s] = \phi(f(\mathbf{z}))_{i,j}[s] \in \mathbb{R}$ using a small positive number $\tau$ as a threshold. We call the SAE latent above the threshold as an *active* latent, otherwise an *inactive* latent. Then we count the number of patches that activate the latent $s$ and consider it as the image-level activation $\mathbf{a}_i[s]$. Similarly, we obtain the class- ($\mathbf{a}_c[s]$) and dataset-level ($\mathbf{a}_D[s]$) activation by incorporating $\mathbf{a}_i[s]$ for the images with the same class or dataset, respectively. Formally, we represent this as below:

$$\mathbf{a}_{i,j}[s] = \mathbb{I}(\mathbf{h}_{i,j}[s] > \tau), \ \text{where} \ 1 \le s \le d_{\text{SAE}}, \tag{4}$$

$$\mathbf{a}_i[s] = \sum_{j=1}^{n_i} \mathbf{a}_{i,j}[s], \ \mathbf{a}_c[s] = \sum_{i \in \mathcal{I}_c} \mathbf{a}_i[s], \ \mathbf{a}_D[s] = \sum_{i \in D} \mathbf{a}_i[s]. \tag{5}$$

From the class- or dataset-level activations, we discover the shared concept within the group. We use these group-wise active SAE latents to analyze the relationship between the interpretable concepts and the model behavior in §4.1.

**Localizing patch-level SAE latent activations.** PatchSAE allows localizing an active latent in the image space. We treat the patch-level latent activation as a soft segmentation mask. Precisely, given an image $\mathbf{x}_i$ and an SAE latent index $s$, we multiply each patch $\mathbf{x}_{i,j}$ with the corresponding latent activation value $\mathbf{h}_{i,j}[s]$ for visualization. For example in Fig. 2(c), we highlight "yellow flowers" or "dogs with black, white, and brown colors" concepts from the input image. Separating the patches relevant to the targeting latent from the input and reference images shows a clearer view of the concept[3]

### 3.3 PATCHSAE DISCOVERS SPATIALLY DISTINCT CONCEPTS IN CLIP

More examples in the following sections are provided in the interactive demo (Fig. 12).

**PatchSAE identifies diverse interpretable concepts.** As depicted in Fig. 3, we explore the SAE latents guided by the statistics. We observe two big clusters of rarely activated with low activation value (bottom left) and frequently activated with high activation (top right), and one small cluster near the center. Although the statistics do not ensure the interpretability of the latents, we find several interesting patterns. Many latents from the bottom left region with high label entropy are

---

[3]We recommend trying the on/off segmentation mask option in the interactive demo.

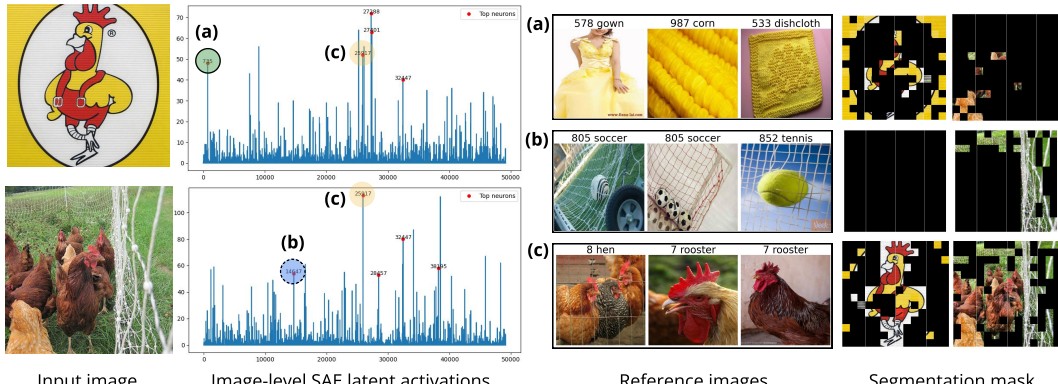

**Figure 4: Localizing SAE latent activations under a covariate shift.** Given two input images of class `hen`, we show image-level aggregated SAE latent activations ($x$-axis: SAE latents index $y$-axis: image-level activation), reference images from ImageNet, and segmentation masks for each input are shown. Among top 10 latents for each input, we pick three interpretable indices where (a) and (b) represent different domains (image style or background) and (c) shows the shared concept.

uninterpretable (Fig. 3(a)). We find more interpretable latents from the second large cluster (top right region). For interpretable latents, lower entropy (Fig. 3(d)) indicates more distinctive semantics such as a specific class, while the higher entropy latent (Fig. 3(c)) represents the shared style of reference images. We also observe multimodal latents that activate when certain text appears in the image. For example, Fig. 3(b) latent detects the text `MILK`. More examples are in Fig. 13.

**PatchSAE provides spatial attribution of concepts.** As a case study, we compare a pair of images having the same class label but different domains (covariate shift) in Fig. 4. The commonly activating SAE latent from both images shows the shared concept `hen` (Fig. 4(c)). From both images, the `hen` latent is activated by the relevant regions. Exclusively activating latents (Fig. 4(c) & (d)) represent discrete concepts such as `yellow` or `net`. The segmentation map highlights the contributing patches for the concepts.

**PatchSAE generalizes across multiple datasets.** Although we train our PatchSAE model only using ImageNet training data, we find that the interpretability of SAE latents is transferrable to different datasets. We show that an SAE latent retrieves a consistent concept from different datasets if such concept exists in the dataset. Otherwise, the mean activation value is low and/or the retrieved images are uninterpretable (§A.3). Fig. 10 shows reference images from ImageNet and four fine-grained datasets for two SAE latents and Fig. 14 shows reference images of top-1 task-wise latent.

## 4 ANALYZING CLIP BEHAVIOR VIA PATCHSAE

In this section, we seek the relationship between SAE latents and model behaviors under classification tasks. By replacing the model's intermediate layer output with SAE reconstructed one and ablating the latents used for the SAE decoder, we find that SAE latents contain class discriminative information (§4.1). Comparing the behaviors of CLIP models before and after adaptation on downstream tasks, we explain the major performance gain stems from adding new mappings at SAE latents to downstream task classes, rather than firing additional class discriminative concepts (§4.2).

### 4.1 IMPACT OF SAE LATENTS ON CLASSIFICATION

We explore the influence of SAE latents on the final model prediction in classification tasks. We replace the intermediate layer representation of CLIP image encoder with the SAE reconstructed output to steer the model output[4] (Templeton, 2024) by selectively using a subset of SAE concepts. Then we compute cosine similarity between text and image encoder outputs for the classification task. Fig. 5(a) summarizes the procedure.

---

[4]We add reconstruction score when replacing the intermediate layer output with SAE reconstructed one following Templeton (2024).

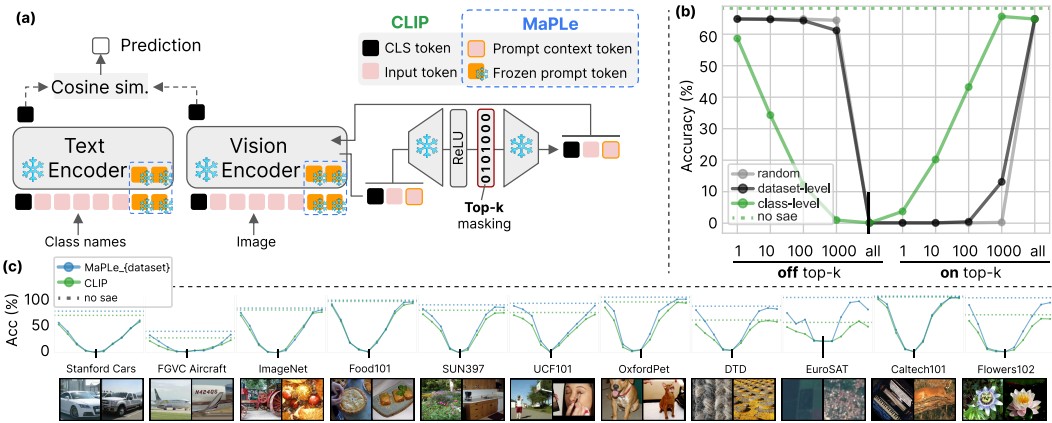

**Figure 5: Top-$k$ SAE latent masking.** **(a)** Top-$k$ SAE latent masking implementation for CLIP and MaPLe. MaPLe adds learnable prompt tokens upon CLIP. **(b)** CLIP (zero-shot) classification accuracy on ImageNet-1K for different SAE latent masking. Top-$k$ selection based on class-level latent activations crucially affects the accuracy while random or dataset-level based selections show marginal or no impact. **(c)** Example images and top-$k$ class-level masking experiment for 11 tasks.

### 4.1.1 ANALYSIS METHOD AND EXPERIMENT SETUP

**Zero-shot classification.** We conduct ImageNet-1K (Deng et al., 2009) zero-shot classification using OpenAI CLIP ViT-B/16 (§Reproducibility) with an ensemble of 80 OpenAI ImageNet templates (Radford et al., 2021) to compute text features for each class and conduct classification by computing cosine similarity between image features out text features (Fig. 5(a)).

**Top-$k$ SAE latent masking.** To select the subset of SAE latents that are used for the linear combination in the SAE decoder, we search for class-wise representative concepts. We utilize *class-level* activations (Eq. 5). In short, we aggregate SAE latent activations from a group of images having the same class label using the training split of each downstream task dataset and find the top-$k$ most frequently activated latents *per class*. We then control the active latents via masking the SAE latent activation vector before feeding it into the decoder. We replace the original model representations with the reconstructed ones by the masked SAE. We compare the classification accuracies by varying the mask. For example, "on top-$k$" refers to using the mask of a one-hot vector where only the top-$k$ latent indices are 1s (active) and the others are 0s (inactive). Contrastingly, "off top-$k$" refers to using a mask filled with 1s except for top-$k$ indices being 0s. As ablation, we provide comparisons on using the same number of *randomly* selected indices and *dataset-level* representative latents (i.e., frequently activated across all classes within the same dataset).

### 4.1.2 KEY FINDINGS

**SAE latents have class discriminative information.** The results for the top-$k$ SAE latent masking experiments are shown in Fig. 5(b,c). Using all SAE latents (on all; identity mask) recovers the original classification performance (i.e., using the original model representation without replacing it with the SAE output) with small reconstruction errors (64.82% for the identity mask and 68.25% for the original). Using the all-zero mask (off all) removes all relevant information, and hence results in the accuracy of 0.1%. Ablating randomly selected or task-wise latents do not show significant affect to the classification accuracy until we use sufficient number of latents. On the other hand, ablating the per-class top activating latents shows a crucial impact on classification performance. We observe a clear performance improvement or degradation in accordance with the increased or decreased number of active SAE latents, respectively. The results of this analysis show that some SAE latents contain rich information that is critical for class discrimination. Moreover, the search for such latents can be narrowed down to the top activating SAE latents that are frequently activated across inputs with the same ground-truth class.

### 4.2 UNDERSTANDING ADAPTATION MECHANISMS

To understand how models are adapted to downstream tasks based on our findings in §4.1, aim to address the following questions: Does adaptation make models *more prominently activate* class

**Table 1: Comparison on class-level SAE latents.** The first six columns show the average number of class-level latents in three groups: high, high-to-low, low-to-high groups per class. Details about three groups are provided in Figure 17. Gray colored for values below 0.1. The next four columns show the accuracy (the same results as dashed lines in Fig. 5(c)) and the final last column shows the performance improvement rate Δ (§ 4.2.1). Rows are sorted by Δ.

| | SAE latents count (average) | | | | | | Accuracy (%) | | | | Δ(%) ↑ |
| | high | | high-to-low | | low-to-high | | CLIP | | MaPLe | | |
| Dataset | base | novel | base | novel | base | novel | base | novel | base | novel | base |
|---|---|---|---|---|---|---|---|---|---|---|---|
| StanfordCar | 11.18 | 11.10 | 0.00 | 0.12 | 0.12 | 0.16 | 53.45 | 66.46 | 56.16 | 60.63 | 5.82 |
| FGVC Aircraft | 10.10 | 10.57 | 0.31 | 0.32 | 0.00 | 0.03 | 21.72 | 28.09 | 31.19 | 30.07 | 12.10 |
| ImageNet | 11.05 | 10.82 | 0.02 | 0.02 | 0.12 | 0.22 | 69.88 | 67.23 | 73.80 | 68.24 | 13.01 |
| Food101 | 11.24 | 11.58 | 0.00 | 0.00 | 0.00 | 0.00 | 84.85 | 86.65 | 87.26 | 89.14 | 15.91 |
| SUN397 | 11.20 | 11.31 | 0.00 | 0.01 | 0.02 | 0.02 | 68.74 | 72.27 | 77.89 | 76.26 | 29.27 |
| UCF101 | 10.90 | 11.10 | 0.01 | 0.01 | 0.07 | 0.06 | 66.56 | 68.80 | 81.13 | 71.40 | 43.57 |
| OxfordPet | 11.11 | 11.73 | 0.00 | 0.00 | 0.24 | 0.16 | 85.41 | 95.26 | 92.04 | 94.99 | 45.44 |
| DTD | 8.85 | 8.96 | 0.32 | 0.36 | 1.68 | 2.02 | 53.12 | 53.44 | 75.17 | 55.62 | 47.03 |
| EuroSAT | 6.90 | 8.70 | 0.50 | 1.00 | 4.10 | 2.30 | 42.61 | 62.42 | 73.07 | 55.77 | 53.08 |
| Caltech101 | 11.25 | 11.32 | 0.02 | 0.00 | 0.00 | 0.00 | 92.57 | 93.47 | 97.49 | 94.29 | 66.22 |
| Flowers102 | 10.81 | 11.47 | 0.01 | 0.00 | 0.01 | 0.01 | 56.32 | 69.26 | 88.07 | 68.41 | 72.69 |

discriminative concepts? Or, do they *define new mappings* between the used concepts and the downstream task classes? The former question refers to the model improving its *perception ability* by adaptation (i.e., adapted models capture additional class discriminative latents). The latter implies that both models recognize similar concepts, but the adaptation *adds new connections* between the activated concepts and the downstream task classes (i.e., adaptation uses concepts that are not closely related to certain classes previously, as class discriminative information).

To answer the questions, we investigate whether the class discriminative SAE latents of zero-shot and adapted models overlap or not. We observe a large overlap between before and after adaptation, which indicates that similar concepts are recognized by the two methods even though the adaptation shows distinctive performance improvement. We thereby conclude our analysis that the major performance gain via prompt-based adaptation stems from tailoring the mapping between (commonly) activated concepts and the downstream task classes.

### 4.2.1 ANALYSIS METHOD AND EXPERIMENT SETUP

**Base-to-novel classification.** Following the setup introduced by Zhou et al. (2022b), we split the downstream task dataset classes into two groups and consider the first half as *base* and the remaining as *novel* classes, then conduct classification on two groups *separately*. We use total 11 benchmark datasets (§Reproducibility): ImageNet-1K (Deng et al., 2009), Caltech101 (Fei-Fei et al., 2004), OxfordPets (Parkhi et al., 2012), StanfordCars (Krause et al., 2013), Flowers102 (Nilsback & Zisserman, 2008), Food101 (Bossard et al., 2014), FGVC Aircraft (Maji et al., 2013), SUN397 (Xiao et al., 2010), DTD (Cimpoi et al., 2014), EuroSAT (Helber et al., 2019), and UCF101 (Soomro, 2012).

**Prompt-based adaptation methods** append learnable tokens to a frozen pretrained CLIP and train the added tokens on downstream tasks. Specifically, MaPLe (Khattak et al., 2023a) adds learnable tokens at the input layer and the first few layers both for text and image encoders (Fig. 5(a)). In the base-to-novel setting, MaPLe uses few-shot samples from each of the base classes to train the learnable tokens. We use officially released MaPLe weights (§Reproducibility) for the experiments.

**Performance improvement via prompt-based methods.** Table 1 (the last four columns) summarizes the reproduced classification results of CLIP and MaPLe in base-to-novel settings. We notice that both zero-shot performance and performance improvement by adaptation vary in a wide range across different datasets. To measure the improvement apart from its zero-shot performance, we compute the improvement rate as (adapted - zero-shot) / (100 - zero-shot) and denote it as Δ. Δ measures the improvement via adaptation relative to the remaining potential improvement from zero-shot performance.

**Transferring SAEs to adapted methods.** Leveraging that prompt-based methods keep model parameters intact as frozen while adding the learned parameters as additional input tokens, we share our PatchSAE trained on the default CLIP model to both CLIP and MaPLe. This allows us to understand the internal mechanisms of the adaptation method by comparing the model behaviors in

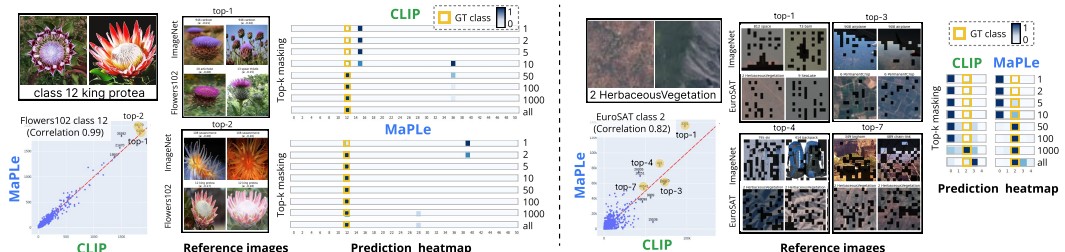

**Figure 6: Different impact of SAE latents.** We show two example cases of Flowers102 (left) and EuroSAT (right). In each figure, we show two example images of the ground-truth (GT) class, a scatter plot view of class-level SAE latent activations comparison ($x$-axis: CLIP, $y$-axis: MaPLe trained on the base classes of the downstream task), reference images from ImageNet and the downstream dataset for top latents, and the top-$k$ masking results as a prediction heatmap. We show reference images with segmentation masks for EuroSAT.

the shared SAE latent space. We demonstrate the transferability of CLIP-based trained PatchSAE to MaPLe by repeating the top-$k$ SAE latent masking experiment with variations to SAE training backbones, SAE latent computing backbones (image encoder), and classification inference backbones (text and image encoder) as CLIP or MaPLe (see §A.4). The results validate the transferability of our PatchSAE to the adapted models under all base-to-novel settings.

**Comparing top SAE latents.** We compute class-level top activating SAE latents using the shared PatchSAE for backbone image encoders CLIP and MaPLe for each task. We plot the comparison of class-level latent activations as a scatter plot (Fig. 6), where each point represents the class-level activation of the latent $s$; $\mathbf{a}_c[s]$ (Eq. 5) with backbones CLIP and MaPLe in $x$ and $y$ axes, respectively. We divide the points into several groups including *high* (high in both), *high-to-low* (high before and low after adaptation), and *low-to-high* (low before and high after adaptation) (Fig. 7). We set the upper and lower bounds using top-50 and top-100 values, and we use class- and dataset-level activations to analyze class- and task-wise performance improvement, respectively.

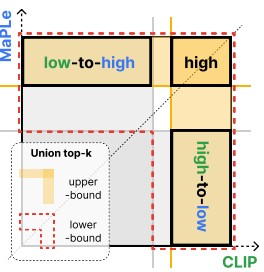

**Figure 7:** Scatter plot regions (see Fig. 17).

### 4.2.2 KEY FINDINGS

**Top activating SAE latents mostly overlap for CLIP and MaPLe.** The first six columns of Table 1 show the normalized count of top activating SAE latents in three groups: high, high-to-low, and low-to-high. For example of EuroSAT base classes, 6.9 latents are highly active in both before and after adaptation, 0.5 previously highly activated latents become non-active after adaptation, and 4.1 previously not active latents became active after adaptation on average. In most cases, SAE latents rarely place in off-diagonal regions. We provide scatter plots for the same comparison in Fig. 20, where we can observe the SAE latent activations are highly correlated. We notice that EuroSAT shows distinctive improvement through MaPLe and the highest count in the low-to-high group and discuss in §B.1.

**Influence of top SAE latents in CLIP and MaPLE are different.** We compare the impact of the same SAE latent for CLIP and MaPLe. Similar to §4.1, we conduct the top-$k$ masking experiment. We provide the results in Fig. 5(c) and deeper case studies in Fig. 6. In Fig. 5(c), regardless of the top-$k$ latent selection backbones, MaPLe consistently show better performance using the same number of SAE latents. The result implies that MaPLe makes a better use of the same (number of) SAE latents for classification than CLIP does. For deeper analysis, we choose two cases in Flowers102 (case 1) and EuroSAT (case 2) that show large performance improvements by adaptation and top-$k$ masking while the former task is a finer-grained classification and the latter is classification in special domains. We observe that zero-shot and adapted models activate SAE latents in similar patterns for images of the same class, while the two models show different predictions. In both cases, top activating latents recognize visual attributes that seem providing representative information relevant ground-truth class. Using this same set of latents, MaPLe makes correct predictions while CLIP does not. This result exemplifies MaPLe adding new connections between commonly activated concepts and the downstream tasks. In essence, our analysis shows that the performance gain of prompt-based adaptation on CLIP can be explained by adding new mappings between the recognized concepts, which do not change much by adaptation, and the downstream task classes.

## 5 DISCUSSION

**Adopting SAEs to vision models.** By adopting SAEs originally studied on LLMs to vision models, this work contributes to the understanding of the vision part of vision-language foundation models. By following basic settings in previous works, evaluating the training performance including reconstruction and sparsity objectives, and performance comparison with the original model in downstream tasks, we validate our design choices. We propose PatchSAE that provides spatial attribution of the candidate concepts, which advances the interpretability. Through extensive qualitative analysis, we demonstrate the interpretability of our SAE. Furthermore, we provide an interactive demo to share abundant results with transparency. The scope of this work focuses on CLIP vision encoder. We believe that the analysis of different vision encoders and extending it to jointly analyzing the multimodality could provide a more comprehensive understanding of large vision-language models. We leave this as future work.

**Understanding adaptation methods.** We use SAEs to shed light on model behaviors and adaptation mechanisms. In order to use the same SAE for both non-adapted and adapted approaches, we focus on prompt-based adaptation method which does not directly update the model parameter but appends learnable tokens as inputs. We choose MaPLe as the prompt-based method because this approach uses learnable tokens in the vision encoder (note that simpler baseline CoOp uses learnable tokens only in text encoder) and shows competitive performance with state-of-the-art methods regardless of its simplicity. Exploring different adaptation methods (e.g., full fine-tuning) as future work could provide deeper insights to adaptation mechanisms of foundation models.

## 6 CONCLUSION

Adapting foundation models to specific tasks has become a standard practice to build machine learning systems. Despite this widespread use, the internal workings of models and adaptation mechanisms to target tasks remains as an open question. To address this question, we introduced PatchSAE, a sparse autoencoder that extracts interpretable concepts with spatial attributions from a database of reference images. We provide a detailed framework to train and analyze PatchSAE models on vision transformers. Through controlled experiments on 11 adaptation tasks, we study how adaptation changes the relation between class outputs and concepts. Surprisingly, our analysis finds that on almost none of the studied tasks, drastically new concepts are introduced for adaptation. Adaptation rather assigns the right *existing* concepts to the correct classes, and in only one task with more notable distribution shift (EuroSAT), we found a non-negligible number of concepts that got suppressed or newly introduced by the adaptation mechanism. Our analysis is an example for leveraging PatchSAE to "debug" inner workings of a vision model on the level of concepts. We believe that methods like PatchSAE will become useful in categorizing algorithms to edit, interpret and adapt foundation models, and to build more effective models on particular downstream tasks.

REPRODUCIBILITY STATEMENT

We used publicly available model checkpoints for CLIP (link) and MaPLe (link). We used OpenAI ImageNet templates for zero-shot classification (link). Code, model weights and raw results are available at `https://github.com/dynamical-inference/patchsae`. We only used publicly available datasets following the official implementation of MaPLe (dataset descriptions) which are cited in the main text.

AUTHOR CONTRIBUTIONS

*Conceptualization:* StS, HsL with comments from JgC; *Methodology:* HsL, StS; *Software:* HsL, JhC; *Formal analysis:* HsL, StS; *Investigation:* HsL, JhC; *Writing–Original Draft:* HsL, JhC; *Writing–Editing:* HsL, StS, JgC.

ACKNOWLEDGMENTS

HsL was supported through a DAAD/NRF fellowship in the NRF Summer Institute Programme, 2024 (57600422). This work was supported by the Helmholtz Association's Initiative and Networking Fund on the HAICORE@KIT partition. This work was supported by Institute for Information & communications Technology Planning & Evaluation (IITP) and National Research Foundation of Korea (NRF) grants funded by the Korea government (MSIT) (RS-2019-II190075, Artificial Intelligence Graduate School Program (KAIST), No. 2022R1A5A7083908, and No. RS-2025-00555621).

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

# A   SAE TRAINING DETAILS AND ABLATION STUDIES

In this section, we provide details about training SAE.

- §A.1 summarizes the training performance of SAEs used for hyperparameter tuning.
- §A.2 shows SAE layer ablation study results.
- §A.3 shows SAE's generalizability to different datasets.
- §A.4 justifies SAE's transferability to adapted models.

## A.1   TRAINING PERFORMANCE

**Quantitative metrics.** We follow the literature on SAEs (Bricken et al., 2023; Templeton, 2024) to set quantitative metrics. The mean squared error (MSE) loss indicates reconstruction ability. Contrastive loss with and without SAE informs the reconstruction ability as well. Close contrastive losses indicate that the SAE reconstructs the input better. The L1 loss and the L0 metric indicate the sparsity of SAE. The lower the values are, the less number of SAE latents are activated for the given input. We start by reproducing training performance as reported in previous studies (Fry, 2024). Then we ablated training hyperparameters $\lambda_{l_1}$, expansion factor, ghost gradient technique[5], and the initialization of decoder bias term (mean vs. geometric median of the training dataset). Furthermore, we ablated model architectures (ViT-B/16 and ViT-L/14), training tokens (CLS vs. all; Fig. 8), and hook layers (§A.2).

**Hyperparameters.** We set the coefficient for L1 regularizer $\lambda_{l_1}$ as 8e-5, the learning rate as 4e-4 with a constant warmup scheduling with warmup step of 500, and initialized decoder bias with geometric median. We train SAE using 2,621,440 samples from ImageNet training dataset using ghost gradient. We set the threshold $\tau$, that we use for transforming patch-level activations into global views (Eq. 4), to 0.2 ($\log 10$ value of -0.7).

**Supplements to summary statistics.** In addition to the frequency and the mean value of activation distribution (Bricken et al., 2023), we use the label entropy (Fry, 2024) and the label standard deviation that can give an intuition about concept granularity. The label standard deviation is tailored for a labeled dataset such as ImageNet, where the label structure contains a hierarchical structure of English words. In this case, the standard deviation indicates whether the latent is capturing a distinct label from ImageNet dataset or other attributes such as the style (or domain) or patterns of image. For example, the dog latent might be activated by different breeds of dogs, so the number of unique labels is high (high entropy) but the gap between labels might be low (low standard deviation). On the other hand, blue color latent might be activated by diverse blue objects or scenes whose labels can be very far away (high entropy and high standard deviation).

Although the quantitative metrics of reconstruction and sparsity validate that SAE is trained as intended, they do not provide rich information about the validity and interpretability of SAEs. Therefore, we utilize SAE latent summary statistics and reference images for qualitative evaluation. We mostly follow the configurations as selected by Fry (2024), confirming that the chosen setup shows reasonable performance. To be compatible with both zero-shot and adapted method MaPLe, which releases official weight on ViT-B/16, we choose model architecture as CLIP ViT-B/16. For a deeper understanding, we use all image tokens in addition to the CLS token. We also note recent progress of SAE architectures and training techniques such as gated SAE (Rajamanoharan et al., 2024), using SAE on other components' output (such as attention output or MLP output), but we focus on the base architecture of SAEs (Bricken et al., 2023) treating the advanced techniques as out-of-scope.

## A.2   SAEs ON DIFFERENT LAYERS

We ablated the model layers to train SAEs. Using ViT-B/16 that has 12 residual block layers, we choose four layers: 2, 5, 8, and 11. We train SAEs for each layers (Fig. 9). **We find that the SAE latents on the deeper layers (i.e., closer to the output) provides semantically richer information with high confidence (high activation value)** than the ones on the shallower layers. Segmentation masks show that top-3 SAE latents represent the semantic of the major object (the Golden Gate

---

[5]Ghost gradient is introduced as an improvement of neuron resampling Bricken et al. (2023) that addresses dead neurons due to sparsity regularization of training.

**Table 2: SAE training configurations and performance.** We share results for six design choices: ViT hooking layer, training tokens, ViT architecture, L1 coefficient $\lambda_{l_1}$, expansion factor ($d_{\text{ViT}} \times$ (expansion factor) = $d_{\text{SAE}}$), using ghost gradient technique, and initialization of decoder bias term. Default configurations are colored as gray. CL indicates the contrastive loss. * indicates baseline result Fry (2024). *Italic* indicates the ablated configuration.

| Layer | Tokens | Arch. | $\lambda_{l_1}$ | Expan. | Ghost | Dec. bias | MSE | CL SAE | CL Org. | L1 | L0 |
|---|---|---|---|---|---|---|---|---|---|---|---|
| 11* | cls* | L/14* | 8e-5* | 64* | T* | geom.* | 0.0027 | 1.775 | 1.895 | 13.900 | 26.00 |
| 11 | cls | L/14 | 8e-5 | 64 | T | geom. | 0.0026 | 1.702 | 1.859 | 13.823 | 29.48 |
| 11 | cls | L/14 | *0.00* | 64 | T | geom. | 0.0000 | 1.926 | 1.927 | N/A | 17570.57 |
| 11 | cls | L/14 | 8e-5 | 64 | T | geom. | 0.0026 | 1.702 | 1.859 | 13.823 | 29.48 |
| 11 | cls | L/14 | *8e-4* | 64 | T | geom. | 0.0069 | 1.693 | 1.856 | 0.077 | 5.57 |
| 11 | cls | L/14 | 8e-5 | *32* | T | geom. | 0.0028 | 1.690 | 1.859 | 13.608 | 35.71 |
| 11 | cls | L/14 | 8e-5 | 64 | T | geom. | 0.0026 | 1.702 | 1.859 | 13.823 | 29.48 |
| 11 | cls | L/14 | 8e-5 | *128* | T | geom. | 0.0025 | 1.689 | 1.816 | 14.506 | 28.43 |
| 11 | cls | L/14 | 8e-5 | 64 | *F* | geom. | 0.0026 | 1.663 | 1.753 | 13.957 | 19.35 |
| 11 | cls | L/14 | 8e-5 | 64 | T | geom. | 0.0026 | 1.702 | 1.859 | 13.823 | 29.48 |
| 11 | cls | L/14 | 8e-5 | 64 | T | *mean* | 0.0027 | 1.693 | 1.859 | 14.050 | 32.47 |
| 11 | cls | L/14 | 8e-5 | 64 | T | geom. | 0.0026 | 1.702 | 1.859 | 13.823 | 29.48 |
| 11 | cls | L/14 | 8e-5 | 64 | T | geom. | 0.0026 | 1.702 | 1.859 | 13.823 | 29.48 |
| 11 | cls | *B/16* | 8e-5 | 64 | T | geom. | 0.0009 | 1.904 | 1.986 | 5.501 | 25.23 |
| 11 | *cls* | B/16 | 8e-5 | 64 | T | geom. | 0.0009 | 1.904 | 1.986 | 5.501 | 25.23 |
| 11 | all | B/16 | 8e-5 | 64 | T | geom. | 0.0025 | 1.885 | 1.962 | 28.300 | 148.56 |
| *8* | all | B/16 | 8e-5 | 64 | T | geom. | 0.0010 | 2.034 | 2.063 | 14.730 | 182.07 |
| *5* | all | B/16 | 8e-5 | 64 | T | geom. | 0.0007 | 2.039 | 2.039 | 10.670 | 298.97 |
| *2* | all | B/16 | 8e-5 | 64 | T | geom. | 0.0005 | - | - | 10.164 | 242.97 |

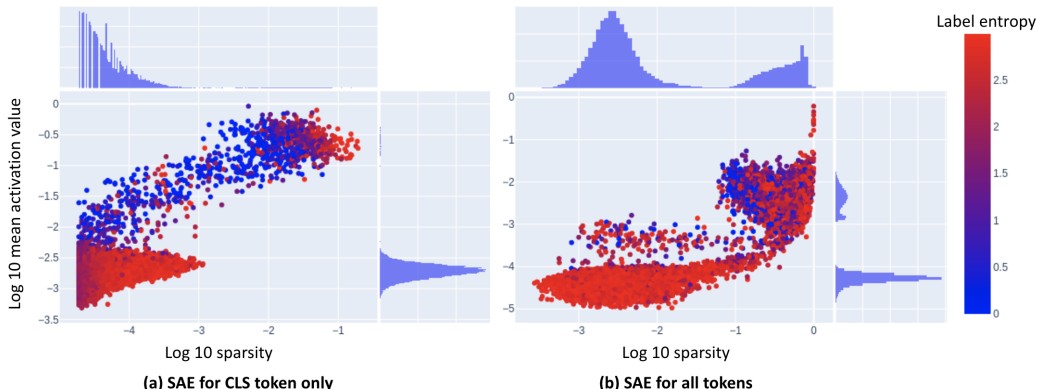

**Figure 8: SAE statistics comparisons on training token ablation.** Training SAE only for (a) the class token and for (b) all image tokens (including the class token) show a difference in statistics. Both have two clusters in the bottom left and top right regions but (b) showed a denser cluster in the top right region than (a). The plotting style is referenced from Fry (2024).

Bridge) in layer 11 while layer 8 and 5 see the triangle shape of the object. Top latents are less interpretable for layer 2. This is unsurprising as the segmentation mask and the activation value indicate the latent is activated by the specified token (local attention), not interpreting a meaningful pattern from the entire semantics. The segmentation mask in reference images separates the concept from the reference images, which enhances the interpretability. We use the same training configuration for all four SAEs.

### A.3 SAE TRANSFERABILTIY TO DIFFERENT DATASETS

In Fig. 10, we show reference images from ImageNet and four fine-grained datasets (Flowers102, Caltech101, OxfordPets, and Food101) for two SAE latents. For the `Christmas` latent (Fig. 10(a)), we retrieve images containing Christmas-related objects or styles. Fig. 10(b) latent represents `hockey` and/or `skate`. From Flowers102 and OxfordPets, the mean activation value

of this latent was low, which explains unclear relationship between reference images and the concept of the latent. The mean activation value is higher in Food101, where the retrieved images are more related to the concept: the left top image shows `hockey` game in the background and the left bottom image shows a game character wearing `roller skates`. The results demonstrate that a SAE latent retrieves a consistent concept from different datasets if such concept exists in the dataset). Otherwise, the mean activation value is low and the retrieved images are uninterpretable. See another example in Fig. 14.

### A.4 SAE TRANSFERABILTIY TO ADAPTED MODELS

To justify using CLIP-based trained SAE for analyzing MaPLe, we repeat top-$k$ SAE latent masking experiment under various settings. We observe consistent results whether using CLIP or MaPLe-based SAE latents and demonstrate the transferability of SAE for multimodal prompt-based adaptation method.

In Fig. 11, we use SAE trained on MaPLe (trained on ImageNet-1K). We compare four settings where classification backbone can be either CLIP or MaPLe and SAE can be either CLIP-based or MaPLe-based trained models.

Fig. 15 and 16 shows top-$k$ SAE latent masking results on 11 datasets. Here, we fix to use class-level activations to select top-$k$ latents and to use CLIP-based trained SAE. We compare four settings of adopting CLIP or MaPLe as SAE activation computing backbone and classification backbone. Using different classification backbone showed different patterns while selecting SAE activation backbone does not show significant difference, which supports the transferability of our SAEs under all base-to-novel settings.

## B ADDITIONAL RESULTS AND SUPPORTING FIGURES

- Fig. 12 shows a screenshot of **interactive demo** short instruction.
- Fig. 13 shows an example of **multimodal** SAE latents.
- Fig. 14 **task-wise** reference images across datasets.
- Fig. 15 and 16 show full results of **top-$k$ SAE latent masking** experiment using MaPLe adapted on each dataset.
- Fig. 17 explains three groups in top activating SAE latent comparison **scatter plots**.
- Fig. 18 and 19 shows **off-diagonal SAE latents** case studies in EuroSAT, DTD and UCF101 datasets.
- Fig. 20 shows the **scatter plot** of aggregated class-level SAE latents in 11 datasets.
- Fig. 21 shows a detailed **confusion matrix** of Flowers102.
- Fig. 22 supplements Fig. 6 and provides more case studies for remapping.

### B.1 TOP ACTIVATING SAE LATENTS

We notice that the EuroSAT dataset shows distinctive performance improvement through MaPLe, low correlation in Fig. 20, and the highest count in the low-to-high group (Table 1). We conduct case studies for classes showing large class-level performance improvement. As shown in Fig. 18, we assess confusion matrices and choose class 2, where the class accuracy improves from 42.61% to 73.07%. We find the class-specific concept latents are found from low-to-high regions (i.e., get activated by adaptation), while concepts irrelevant to the classes are deactivated (high-to-low). We find concepts that are generally related to the task (satellite or pictures from an airplane) in the high (diagonal) group. We provide more case study results in Fig. 19. The results yield positive initial findings for the first research question – *adaptation activates additional class-related latents* – and suggest the need to examine the possibility of adaptation improving perceptual ability. We leave this as future work.

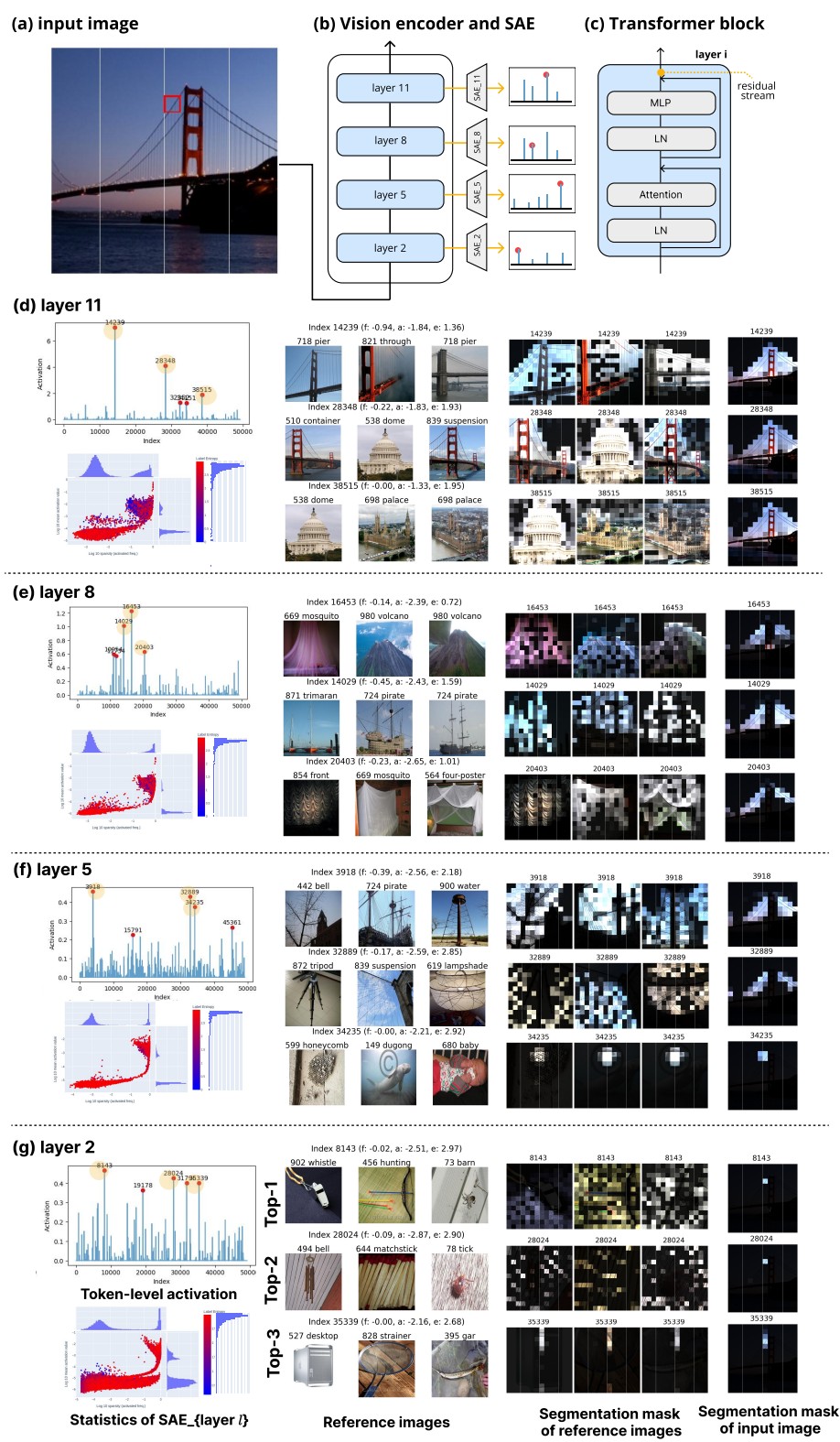

**Figure 9: SAEs on different layers.** We pass an **(a)** input image to **(b)** a vision transformer and collect **(c)** residual stream output from layers 2, 5, 8, and 11. **(d-g)** shows the token-level SAE latent activations ($x$-axis: SAE latents index $y$-axis: activation value) from the image token at the patch highlighted in **(a)**, (left), reference images and segmentation mask of top-3 latents (middle), and the summary statistics of the corresponding SAE (right) for each layer.

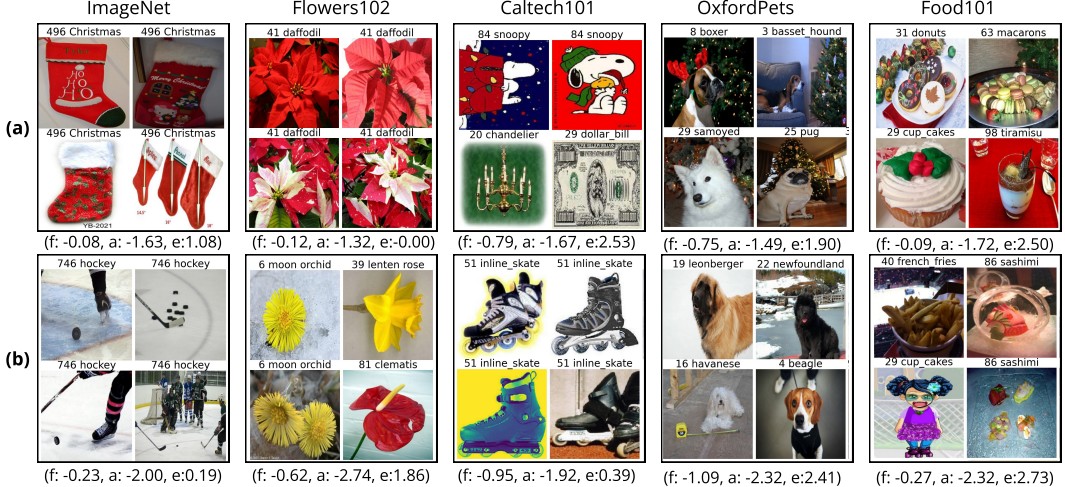

**Figure 10: SAE latents are generalizable to different datasets.** Reference images of two SAE latents **(a)** (top) and **(b)** (bottom) from five datasets. We present label and class name above each image. The latent statistics log10 of activated frequency, log10 of mean activation, and label entropy values computed from each dataset are summarized as (f, a, e) below four reference images. More examples are shown in the interactive demo.

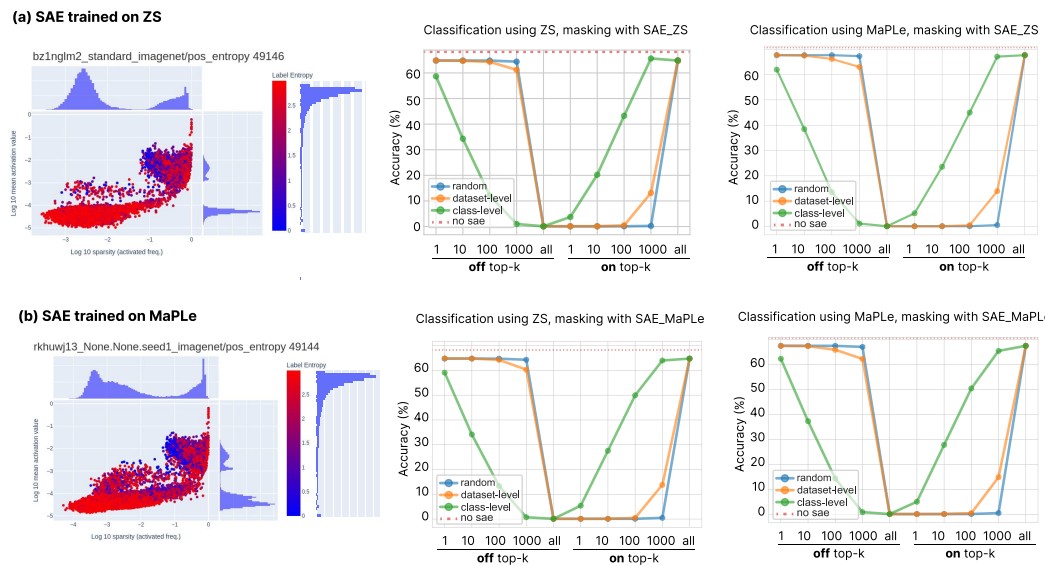

**Figure 11:** Comparison on SAEs trained on (a) CLIP (zero-shot) and (b) MaPLe (adapted on ImageNet-1K dataset) models. Left: The scatter plot shows the summary statistics of SAE latents. Right: We repeat the SAE latent masking experiment done in §4.1 for four combinations of varying the classification model and the SAE model between CLIP and MaPLe. The top left plot is the same plot from Fig. 5(b).

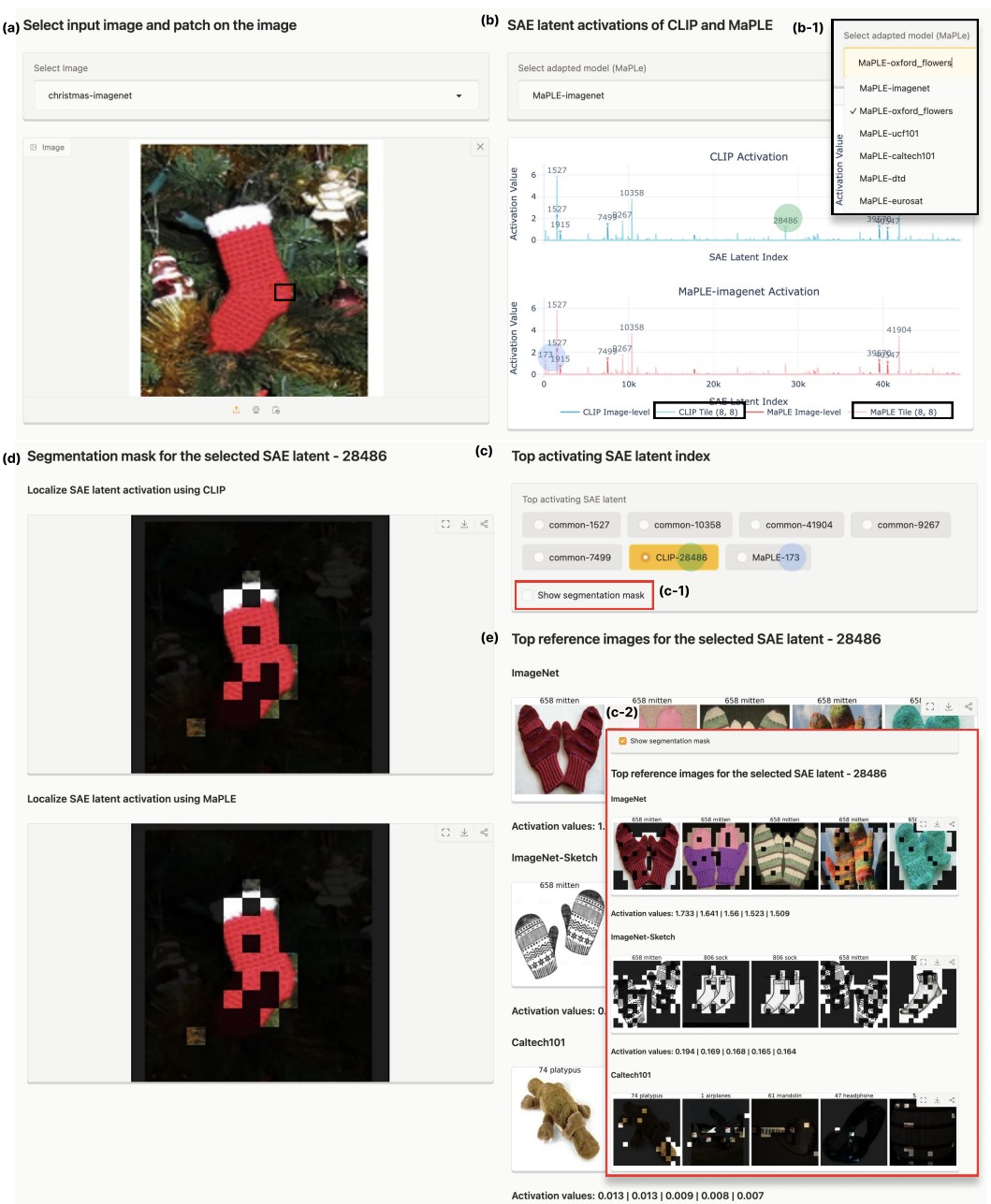

**Figure 12:** interactive demo. (a) Select input image. Specifying patch is also available. (b) Select image encoder backbone for SAE latents. We provide CLIP as default and MaPLe trained on different datasets for comparisons. We show image-level and patch-level (if a patch is specified) SAE latent activations. (c) Top SAE latents (commonly / only in CLIP / only in MaPLe) are selectable. We show (d) segmentation mask and (e) reference images (and activation value of each image) for the selected index. We provide (c-1) on/off option for segmentation mask in reference images. (c-2) shows reference images with segmentation mask.

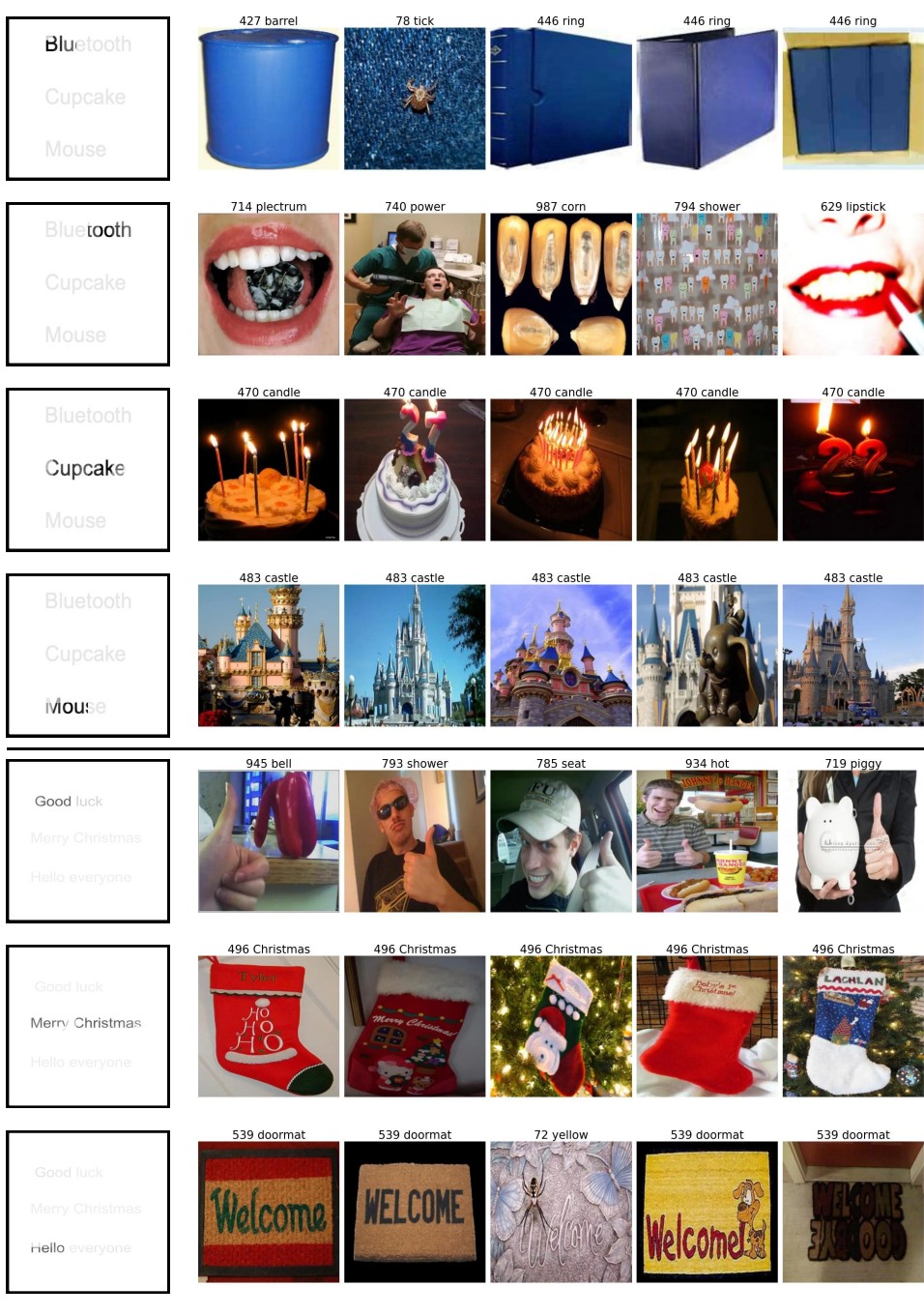

**Figure 13: SAE latents can be multimodal**. We find multimodal SAE latents that activate from both text and images of the same concept. Each row shows different SAE latent. The first column shows the segmenation mask from the input image and the right five columns are reference images. From top to bottom, each latent represents blue, tooth, (cup)cake, mouse (as a representative animation character), good, Christmas, and greetings (hello and welcome). SAE latent activation value and more examples are provided in the interactive demo.

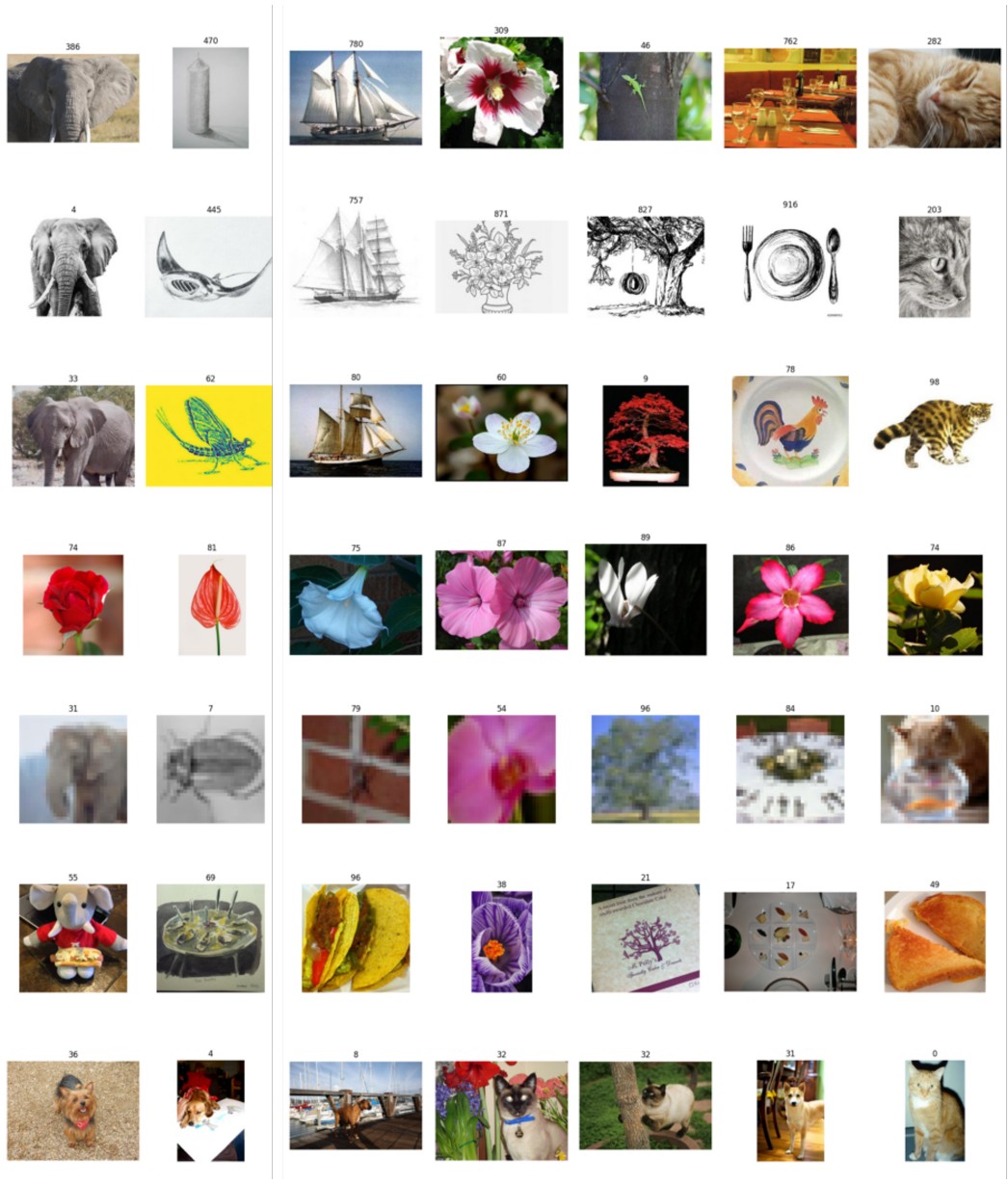

**Figure 14: Reference images for task-wise most representative SAE feature.** $(i, j)$-th element indicates the reference image from $j$-th dataset for $i$-th task-wise feature. From left to right (same order from top to bottom), datasets are ImageNet, Imagenet-Sktech, Caltech101, Flowers102, CIFAR100, Food101, and OxfordPet.

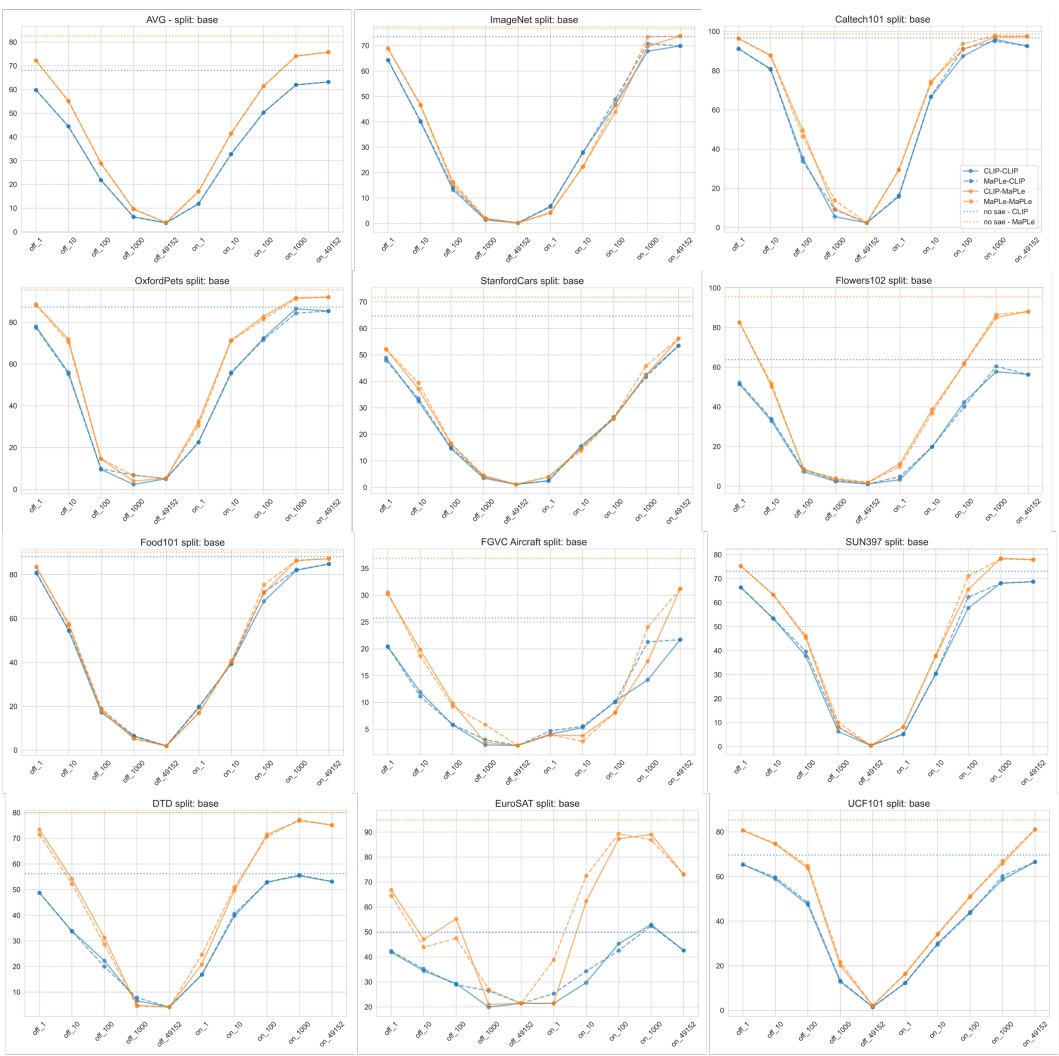

**Figure 15: SAE feature masking on base-to-novel classification task (Base split).** The legend indicates (backbone for SAE latent selection)-(backbone for classification inference). Blue and orange colors for CLIP and MaPLe as classification inference backbones, respectively.

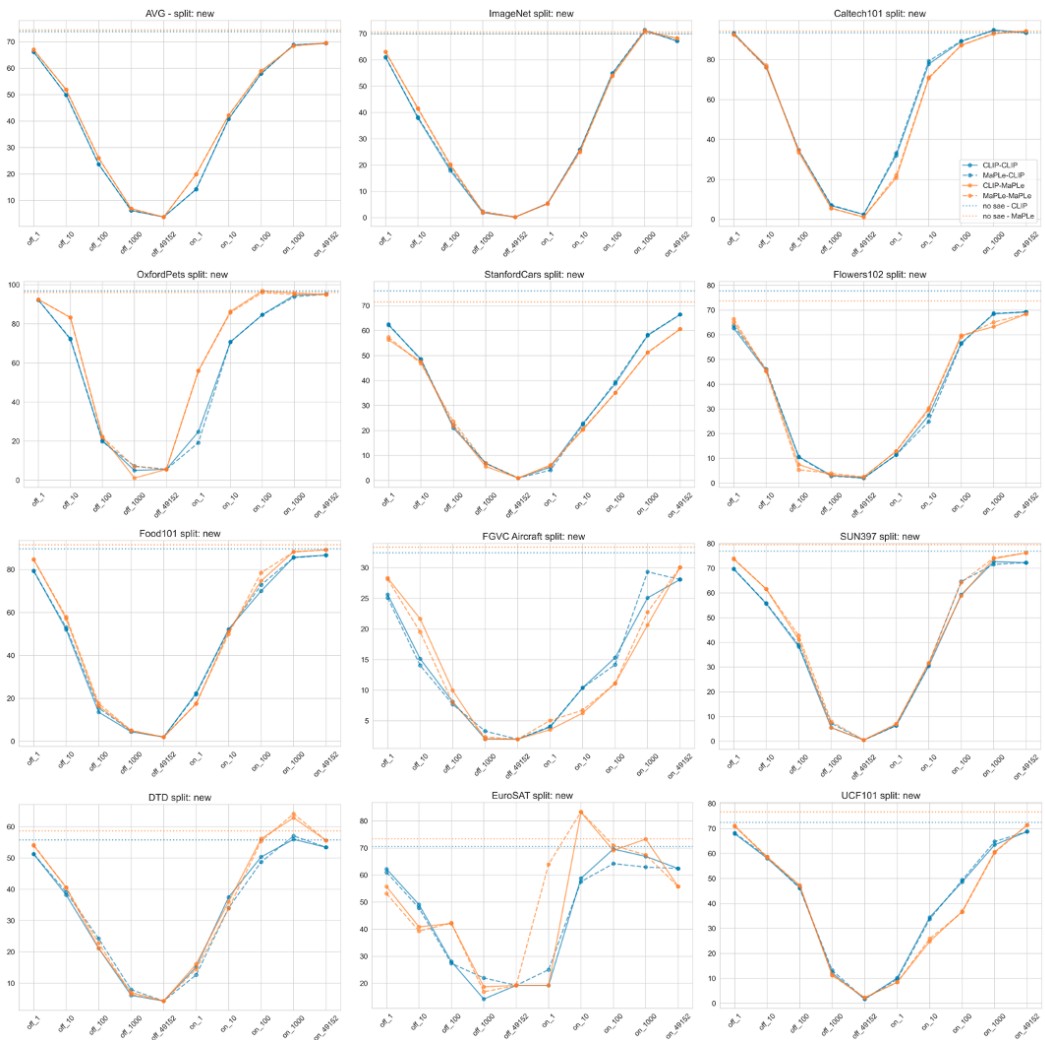

Figure 16: **SAE feature masking on base-to-novel classification task (Novel split).**

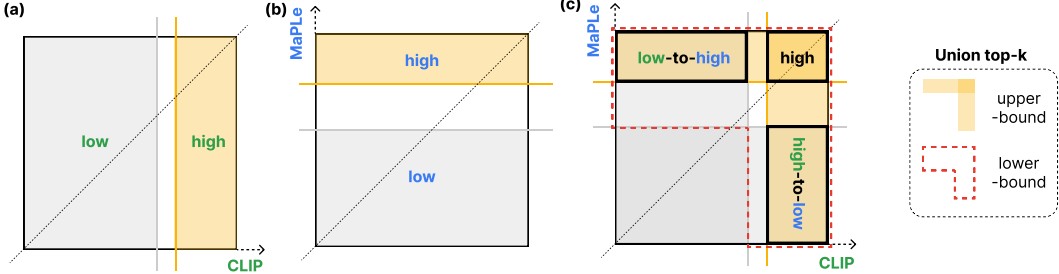

Figure 17: **SAE latent comparison scatter plot.** We compare group (class or dataset) level SAE latents of two backbone image encoders CLIP and MaPLe and plot it as a scatter plot. Each point represents group-level activation (Eq. 5) of CLIP ($x$-axis) and MaPLe ($y$-axis). We set upper and lower bounds using union top-50 and top-100, respectively. We focus on three groups: high (high in both), high-to-low (high before and low after adaptation), and low-to-high (low before and high after adaptation).

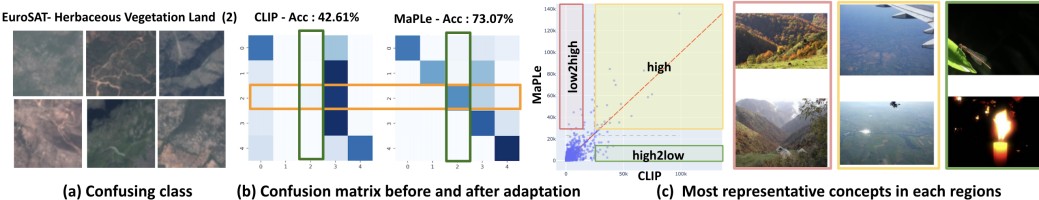

**Figure 18: Case study on EuroSAT.** (a) Images of classes with low classification performance in the CLIP model. (b) After adaptation, classification performance improves for these classes. (c) Representative images of features that show distinct activation patterns: those that were highly activated before adaptation but decreased afterward (high-to-low), those that had low activation before adaptation but increased afterward (low-to-high), and those that remained highly activated both before and after adaptation.

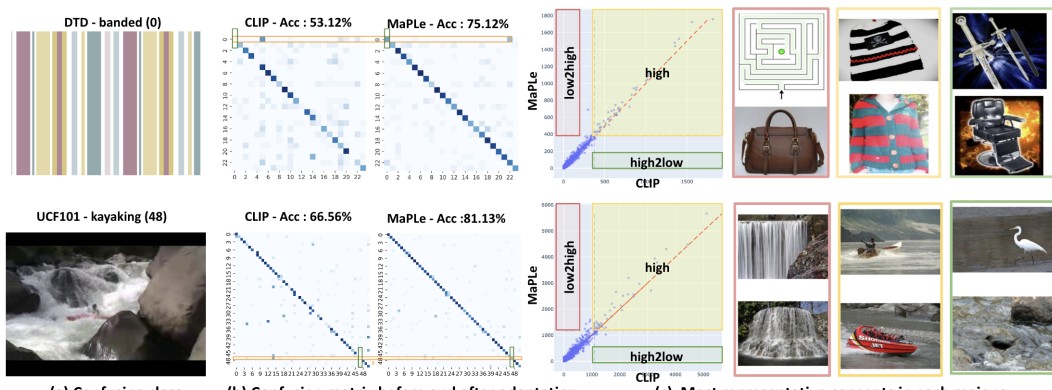

**Figure 19: Class-level SAE latents in three regions on DTD and UCF101.** We show reference images of highly activated before adaptation but decreased afterward (high-to-low), latents with low initial activation that increased after adaptation (low-t0-high), and latents that remained consistently highly activated both before and after adaptation (high).

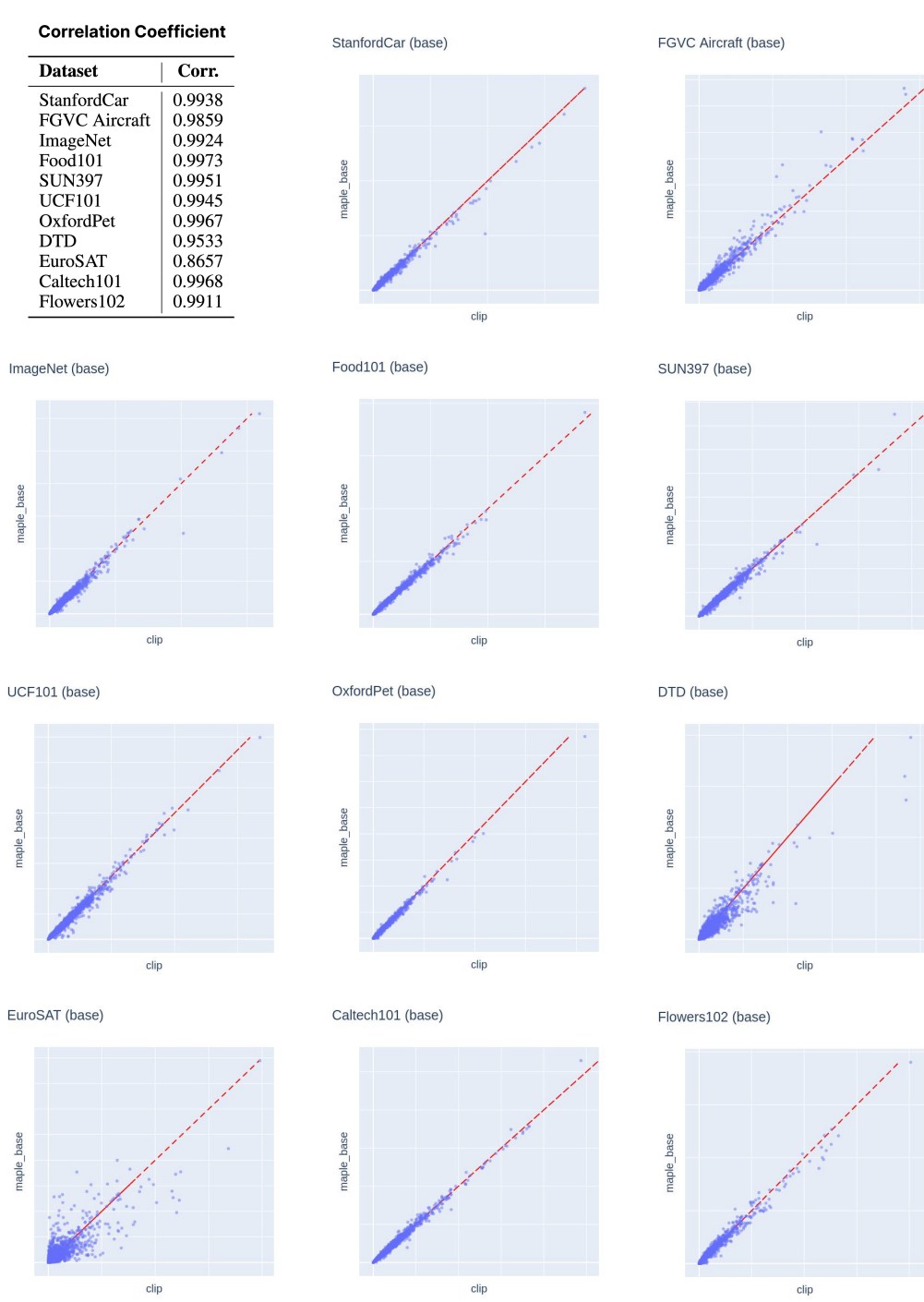

**Correlation Coefficient**

| Dataset | Corr. |
| --- | --- |
| StanfordCar | 0.9938 |
| FGVC Aircraft | 0.9859 |
| ImageNet | 0.9924 |
| Food101 | 0.9973 |
| SUN397 | 0.9951 |
| UCF101 | 0.9945 |
| OxfordPet | 0.9967 |
| DTD | 0.9533 |
| EuroSAT | 0.8657 |
| Caltech101 | 0.9968 |
| Flowers102 | 0.9911 |

**Figure 20: Dataset-level top activating SAE latents.** Correlation coefficient values and the scatter plot view on base-to-novel classification datasets base split. Top activating SAE latents for CLIP and MaPLe are highly correlated.

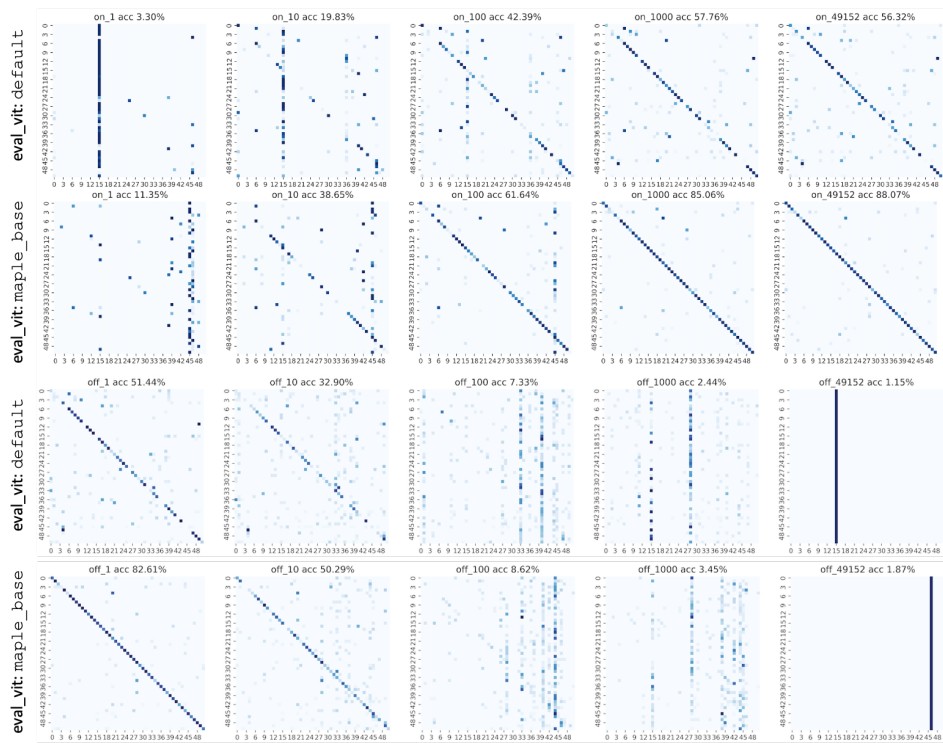

Figure 21: Confusion matrix for Flowers102 dataset

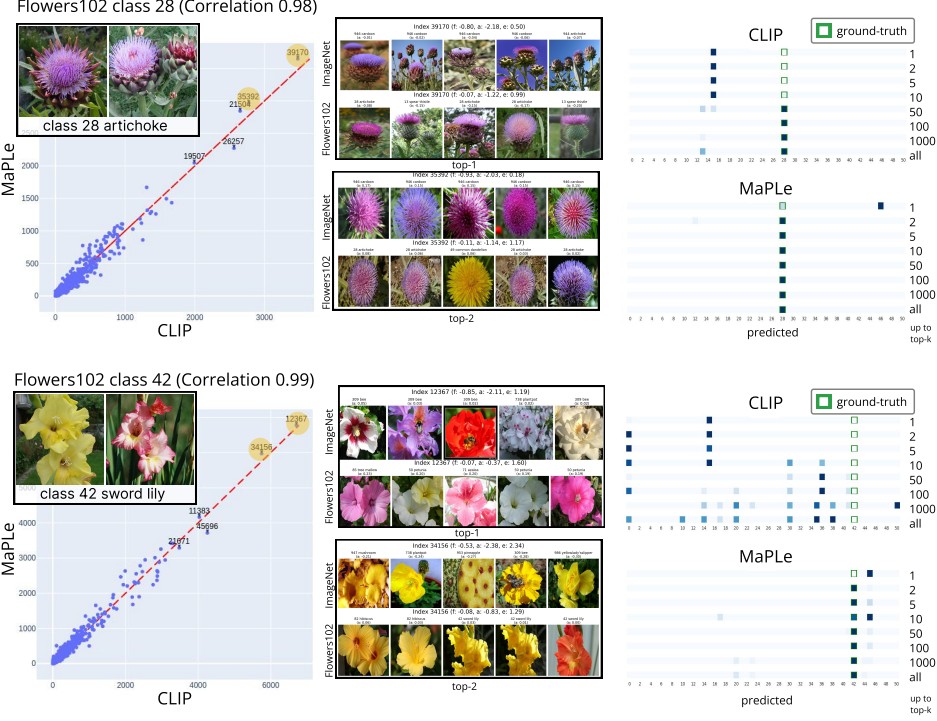

Figure 22: **Supplements for Fig. 6.** We show example cases where SAE latents with class discriminative information are highly activated in both models, but only the adapted model makes correct predictions (when using top 2 latents).

