# OpenReview forum: "Sparse autoencoders reveal selective remapping of visual concepts during adaptation"
_ICLR.cc/2025/Conference — ICLR 2025 Poster_

### Official Review · Reviewer_h3c2 · 2024-10-28

**Soundness:** 3
**Presentation:** 2
**Contribution:** 3
**Rating:** 6
**Confidence:** 4

**Summary:**

This paper utilizes sparse autoencoders (SAE) to interpret visual concepts learned by the CLIP vision model and investigates how these concepts are affected by adaptation techniques such as MaPLe. The key finding is that during adaptation, the model primarily reuses existing concepts rather than learning entirely new ones.

**Strengths:**

- The paper offers a detailed and fine-grained exploration of CLIP by processing the entire token sequence with SAEs, allowing for a deeper understanding of the model's visual feature representations.
- It tackles the adaptation dynamics of foundation models like CLIP, which is a relatively under-explored topic.

**Weaknesses:**

- The clarity and organization of the analyses are insufficient. The main body of the paper frequently refers to figures in the appendix, many of which are poorly annotated, making it difficult for readers to follow the core arguments. The authors should revise the manuscript to consolidate the most important findings in the main body and present them in a clearer and more structured format.
- The paper lacks sufficient discussion on the broader implications and significance of its findings. Are any results surprising or do they challenge conventional views within the field? The authors should also elaborate on the practical value of their discoveries.

**Questions:**

- Section 3.3 (ADAPTATION METHODS AND SAE) is difficult to follow. Can you provide a more concise and coherent explanation of the logic behind the analysis, supported by a smaller set of essential figures?
- What does “suddenly capture new concepts” mean in this context? In human cognition, new concepts are usually built upon existing knowledge. Given this, how does this study distinguish between the reuse of old concepts and the acquisition of new concepts?

---

> ### Author Response · Authors · 2024-11-21
> **Response to Reviewer h3c2**
>
> *We appreciate the considerate reviews and raising important questions. We hope our responses address the reviewer's concerns.*
>
> ### **h3c2-W1**
> > The authors should revise the manuscript to consolidate the most important findings in the main body and present them in a clearer and more structured format.
>
> We revised the overall structure of the paper. We added **separate subsections for key findings** in each section in $\S$ 3 and $\S$ 4. Please see **GR1**.
>
> ### **h3c2-W2**
> > The paper lacks sufficient discussion on the broader implications and significance of its findings.
>
> **We clarified contributions and added broader implications in $\S$ 5** and in overall sections. Please see **GR3**.
>
> Important findings and their practical value are as follows:
> 1. SAE identifies diverse interpretable concepts which inform spatial locality ($\S$ 3.3).
>     * Providing which group of patches (i.e., segmentation mask) activates certain SAE latent **enhances the interpretability** of our SAE.
> 2. There are SAE latents that have a crucial impact on class discrimination ($\S$ 4.1.2).
>     * Findings of the strong connection between interpretable concepts and model behavior **"opens up more avenues on mechanistic interpretability research"** (quoting GMQj Strength 2).
> 3. While CLIP and MaPLe activate SAE latents in similar patterns, the impact of the same latent in two models is different ($\S$ 4.1.2).
>     * Our analysis provides initial results for an important problem -- **understanding the adaptation** mechanism of the vision encoder which is a core visual component in large vision-language models, such as LLaVA-1.5.
>
> ### **h3c2-Q1**
> > Can you provide a more concise and coherent explanation of the logic behind the analysis, supported by a smaller set of essential figures for "Adaptation methods and SAE"?
>
> **We revised $\S$ 4.2 (previous $\S$ 3.3)** to clarify the terms used for explaining the adaptation mechanism. **We added Fig. 6** which illustrates the analysis.
>
> In short, we find that both non-adapted and adapted models recognize **similar concepts** (supported by the high correlation in the SAE latent activation scatter plot). However, the **model predictions vary** although they use the same concepts (supported by different patterns in the prediction heatmap).
>
> Please see **GR1** for overall changes made to address the ambiguities.
>
> ### **h3c2-Q2**
> > What does “suddenly capture new concepts” mean in this context? In human cognition, new concepts are usually built upon existing knowledge. Given this, how does this study distinguish between the reuse of old concepts and the acquisition of new concepts?
>
> We consider the top activating latents as recognized (captured) concepts from an input. To analyze the model "using" the concept, we conduct the top-k masking experiment and observe the model outputs (Fig. 6). In the experiment, we control the active concepts used for SAE reconstructing the model representation and replace the reconstructed output with the original model representation.
>
> **We compare the top activating latents** in before (CLIP) and after (MaPLe) adaptation from the same input to analyze the *acquisition of new concepts* ($\S$ 4.2.2). We force two models to use the same concept in the SAE decoder and **compare the model predictions** (i.e., the *"use" of the concepts*) ($\S$ 4.2.2).
>
> **We revised regarding expressions and added explanations in the paper $\S$ 4.2.2** to convey clearer messages.

---

> > ### Author Response · Authors · 2024-11-24
> >
> > Dear reviewer, given that the end of the discussion period is approaching, we wanted to reach out and discuss any further questions you might have. We are happy to address further follow-up questions, in particular around the broader implications of our findings. In any case, thank you for taking the time to consider our rebuttal and the substantial changes we made to the manuscript. Your comments helped a lot for preparing this revision.

---

> > ### Comment · Reviewer_h3c2 · 2024-11-25
> >
> > I have reviewed all the rebuttals and greatly appreciate the author's efforts in revising the entire manuscript. Based on these improvements, I will be increasing my score accordingly.

---

> ### Author Response · Authors · 2024-11-30
>
> Thank you for your favorable evaluation of our revised manuscript. If you have any further questions or points requiring clarification, please let us know.

---

### Official Review · Reviewer_qx77 · 2024-10-29

**Soundness:** 3
**Presentation:** 2
**Contribution:** 2
**Rating:** 6
**Confidence:** 2

**Summary:**

In this paper, the authors propose a new way to try to interpret features learned from a vision-language (in particular, a CLIP model). The proposed approach train SAE on all image tokens. The author then show that the trained SAE can be used to discover different concepts from the CLIP vision transformer. The authors then investigate how the learned "interpretable features" relates the output of the model.

**Strengths:**

- Better understanding foundantion models (and what their features encode) is an extremely important problem---specially as these models get more powerful.
- The proposed approach is simple  and leverages the well-understood sparse autoencoder approach to interpret the features.

**Weaknesses:**

- The paper has some very strong claims that has not been properly demonstrated. Eg, L249 says "CLIP understands sophisticated concepts from input images not only responding to basic patterns such as color or shape ". Showing a a few (potentially cherrypicked) qualitative examples is not enough to show/demonstrate/prove the results on the paper. It would make results stronger if  the authors could provide more quantitative metrics to measure the sophistication of detected concepts. This would make the results on the paper more convincing/interesting.
- Some sessions of the paper are not very clear. For example, the authors leverage MaPLe model (eg "adaptation methods analysis") and nowhere in the paper they explain how this model works. It would make the paper more readable if the authors could provide a brief overview of this model on the manuscript.
- A lot of design choices seem to be made ad-hoc. For example, why chose the second-to-last layer? Why use the SAE feature statistics used? Why 49,152 dimensions? Could the authors provide some justification for these choices? How were those choices made? Was any ablation study performed to choose those values?

**Questions:**

- Please see "weaknesses" above for a few questions.
- How would the results change if the authors use a subset (or a totally different set) of "SAE feature statisics"? Why did the authors chose the ones they use in the paper (eg, sparsity, label entropy, mean activation, etc). Was any ablation study performed to select these features? Were they selected in previous work?
- How would the proposed approach perform on other language-vision models besides CLIP? Would the selected hyperparameters need to be different in this case?

---

> ### Author Response · Authors · 2024-11-21
> **Response to Reviewer qx77**
>
> *We appreciate the constructive feedback and positive support. We hope our responses address the reviewer's concerns.*
>
> ### **qx77-W1**
> > It would make results stronger if the authors could provide more quantitative metrics to measure the sophistication of detected concepts.
>
> We use the label entropy as a proxy for the "abstractive level" of SAE latents. We add Fig. 3 which shows SAE latents with different statistics. We also use the label standard deviation to sort the latents and see relative abstractiveness.
>
> Although the label entropy informs if one latent represents a specific class or not, these statistics are incomplete to rigorously define the granularity of SAE concepts as the reviewer pointed out. **We revised the expression** "CLIP understanding sophisticated concepts" to "SAE latents represent a wide range of interpretable concepts".
>
> To address the concerns regarding cherrypicking of the results, we provide an **[interactive demonstration](https://huggingface.co/spaces/iclranonym/paper14240)** which presents **more qualitative results**. We plan to add more examples.
>
> ### **qx77-W2**
> > It would make the paper more readable if the authors could provide a brief overview of MaPLe on the manuscript.
>
> We revised the overall manuscript to improve readability and clearly provide the necessary details. We add a separate section for the analysis method and experiment setup before the key findings of each section, **including an explanation of MaPLe ($\S$ 4.2.1. and Fig. 5(a))**. Please see **GR1** for the summary of the overall paper revision.
>
> ### **qx77-W3 & Q1**
> > Could the authors provide some justification for design choices (including ViT layer, feature statistics, and SAE dimensions)?
>
> We conducted the ablation studies to set hyperparameters, but as the reviewer pointed out, we have reported a limited part of the results in the previous manuscript. **We added quantitative results** used for hyperparameter tunings (including SAE dimensions) in $\S$ A.1. Please see **GR2.2**. We state the justification for **ViT layer** in **GR2.3**.
>
> We utilize the activated frequency, the mean activation value, and the label entropy as summary statistics following previous work (Pricken et al., Fry et al.). We additionally use label standard deviation, which is tailored for special cases like ImageNet where close labels are semantically close to each other. **We added the references** for choosing the statistics in the manuscript ($\S$ 3.2).
>
> ### **qx77-Q2**
> > How would the proposed approach perform on other language-vision models besides CLIP? Would the selected hyperparameters need to be different in this case?
>
> We believe our proposed PatchSAE and analysis methods are applicable to other vision encoders with tailored modifications and optimal hyperparameter tuning. In our view, the current set of training configurations would provide a good starting point for training SAEs on different vision models. Since SAEs on vision models are relatively under-explored, we answer based on SAEs for various LLMs. We plan to examine this experimentally in future studies.

---

> > ### Author Response · Authors · 2024-11-24
> >
> > Dear reviewer, we would be very happy to discuss any further questions before the end of the discussion phase. In particular, please let us know if your questions around the design choices behind our method, and the quantitative evaluation have been resolved by our edits. For instance, we find the results of the new layer-wise analysis (Figure 9) to motivate the choice for the selected layer of main analysis quite interesting.
> >
> > Thanks again for your comments which helped a lot for revising our manuscript.

---

> > > ### Comment · Reviewer_qx77 · 2024-11-24
> > >
> > > I thank the authors for their rebuttal and the revised version of the manuscript. Some of my concerns were addressed and I raise my score.

---

> > > > ### Author Response · Authors · 2024-11-25
> > > > **Thank you!**
> > > >
> > > > Dear reviewer, thanks a lot for re-considering your initial assessment of the paper after our rebuttal. Can we address any remaining questions, or generally further strengthen the paper in the remainder of the rebuttal phase? Any open questions?
> > > >
> > > > Thanks again for the comments you already provided, they were very helpful to get the manuscript into its current shape, and motivated many of the new experimental results we added. If you feel that some of them are not sufficiently addressed, please let us know.

---

### Official Review · Reviewer_67Ef · 2024-11-10

**Soundness:** 3
**Presentation:** 2
**Contribution:** 2
**Rating:** 6
**Confidence:** 4

**Summary:**

This paper introduced Sparse AutoEncoder (SAE), which previously trained to disentagle and interprete LLM hidden states, to analysis CLIP features. The authors trained a SAE on top of CLIP features with ImageNet samples, and carried out several experiments with it demonstrating 3 points:
- Section 3.1: certain demension in SAE feature correlates to certain visual concepts
- Section 3.2: targeted ablation of neurons yeilds degraded performance
- Section 3.3: CLIP model "re-use already known concepts" when it is fine-tuned for downstream datasets

**Strengths:**

- The paper transfers the methodology of SAE from NLP to vision. The idea of identifying visual concepts from CLIP features is quite interesting.
- From visualizations, it seems that the SAE on top of CLIP sucessfully identified some meaningful semantic concertps in both image-level and patch-level.

**Weaknesses:**

1. **Lacks quantitative evaluation of SAE's reliability**. Although SAEs for LLMs are widely studied in NLP, to my best knowledge, the usage of SAE for CLIP is only reported in several blog posts that have not been peer-reviewed. Since its effectiveness and reliability are not sufficiently justified in a convincing way, directly using it to draw conclusions is quite risky.

    - For example, how can we ensure that the training settings listed in Section 4.1 are properly set? Are 49,152 hidden dimensions enough for the vision domain? Does this SAE achieve monosemanticity?

    - Examples in the case study are certainly not sufficient for a rigorous evaluation. I would suggest conducting a large-scale evaluation using vision datasets with fine-grained attribute annotations to answer the above questions.


1. **Potential distribution shift between base and fine-tuned CLIP**. As stated in Line 325, the authors used the SAE trained on top of pretrained CLIP to interpret the fine-tuned CLIP. As MaPLe fine-tunes both image and text encoders, there are no guarantees that the feature distribution will remain the same after fine-tuning, and all the conclusions in Section 3.3 might be inaccurate as a result.


1. **Regarding "re-mapping" or "re-using" visual concepts**. As highlighted in the paper title and section titles, the authors wanted to demonstrate the connection of visual concepts between base and fine-tuned CLIP. However, I am a bit confused regarding the terminology used, as neither "re-mapping" nor "re-using" is properly defined in the paper. The authors are encouraged to provide a clear definition of these terms and highlight how the results (e.g., Fig. 6c) demonstrate this point.


1. **Significance of Contribution**. Overall, I find the new insights provided in this paper are not as substantial as I expected when I read only the title. Training SAE on CLIP features is not novel. It has been previously reported in several blog posts, yet this paper does not provide a more formal and quantitative evaluation. The results in Section 3.1 and 3.2 are somewhat expected and are not particularly surprising.

    I would suggest that the authors conduct large-scale quantitative validation to prove the effectiveness of CLIP's SAE, e.g., ablation of hyperparameters, training settings, and types of CLIP models (convolutional encoders are not covered in the presented study). Additionally, clearly stating the relation and differences between SAE and existing interpretability methods in the vision domain would help readers understand the contributions.

**Questions:**

See weaknecess.

---

> ### Author Response · Authors · 2024-11-21
> **Response to Reviewer 67Ef**
>
> *We appreciate the constructive and incisive feedback. We hope our responses address the reviewer's concerns.*
>
> ### **67Ef-W1**
> > Lacks quantitative evaluation of SAE's reliability.
>
> **We added evaluation setup and quantitative results** to validate SAE's reliability. Please see **GR2.1**.
>
> As the reviewer pointed out, training SAEs on vision models is under-explored. Following previous work (Templeton et al., Bricken et al.), we evaluate the reconstruction ability and the sparsity using training metrics such as MSE loss and L1 loss. We set the training configurations through ablation studies (Table 2). When no significant difference is observed, we follow the selection in existing work (Fry et al.).
>
> The SAE dimension (the expansion factor) does not show a significant difference in terms of training metrics. Although the sparsity (L1 and L0) metrics can be used as proxies of monosemanticity, the metrics are insufficient to clearly evaluate if the SAE achieves monosemanticity. We thereby stick to using the value selected by the baseline. We lower the tone that emphasizes the absolute value of the SAE dimension size in Abstract.
>
> ### **67Ef-W2**
> > Potential distribution shift between base and fine-tuned CLIP.
>
> We appreciate the reviewer pointing out an important point to strengthen the reliability of our analysis. To minimize the risk of distribution shift, we choose prompt-based adaptation methods that do not update the model parameters but append learnable tokens as additional inputs. **We demonstrate the transferability of SAE** in both methods ($\S$ 4.1.1, $\S$ A.4, and Fig. 11). Please see **GR2.4**.
>
> ### **67Ef-W3**
> > Regarding "re-mapping" or "re-using" visual concepts.
>
> **We revised $\S$ 4.2 and added Fig. 6** to clarify explaining adaptation mechanisms. Please see **GR1** for overall changes made to address the ambiguities.
>
> The key logic behind $\S$ 4.2 is that: We find that both non-adapted and adapted models recognize **similar concepts** (supported by the high correlation in the SAE latent activation scatter plot). However, the **model predictions vary** although they use the same concepts (supported by different patterns in the prediction heatmap). Therefore, we conclude that the adaptation adds new mappings (re-map) between the commonly fired concepts and downstream task classes.
>
> ### **67Ef-W4**
> > Significance of contribution.
>
> We clearly state the distinctive contribution in $\S$ 2 and $\S$ 5. Please see **GR3**.
>
> In short, the main contributions of this work are as follows:
> 1. We share the full framework of training SAE and using it as an interpretability on vision transformer. Distinct from previous works, we propose to use **patch-level** image tokens for SAE latent analysis which provides **spatially localized attributions** of SAE latents and is easily transformable to a higher (image- / class- / dataset-) level of analysis.
> 2. Using SAE as an interpretability tool, we examine the relationship between i**nterpretable concepts** and **downstream task-solving ability** in the CLIP vision encoder. This analysis addresses an important problem--understanding the **adaptation mechanisms** of a core vision component of large vision-language foundation models.

---

> > ### Comment · Reviewer_67Ef · 2024-11-22
> > **Thanks for the detailed response**
> >
> > The revisions addressed my concerns. I feel the current version is much more complete now. I have raised my score.

---

> > > ### Author Response · Authors · 2024-11-23
> > >
> > > Thanks a lot for the positive assessment of our revised manuscript. Please let us know in case you have further questions we can address.

---

### Official Review · Reviewer_GMQj · 2024-11-11

**Soundness:** 3
**Presentation:** 4
**Contribution:** 2
**Rating:** 8
**Confidence:** 4

**Summary:**

The authors perform sparse autoencoder (SAE) training on the features of a pre-trained vision transfer (ViT) from the CLIP image encoder.. The outputs of the 11th layer of the CLIP ViT is used to train a two layer SAE with reconstruction + sparsity objective. The SAE encoder features are then used to validate the presence of concepts in CLIP representations, and their role in classification as well as downstream tasks. The authors first utilize feature and example based statistics to sanity check that concepts are indeed present in SAEs trained on the CLIP visual representations, as well as multi-modal CLIP representations. Next, they verify that these conceptual features are relevant for class discriminative performance of CLIP zero-shot classification. Lastly, the authors show that SAE conceptual features after prompt-based adaptation show a suppression in non class-relevant features, while and increase in the activation of class-relevant features. New conceptual features are not learned during this adaptation procedure.

**Strengths:**

* The authors do a good job at utilizing both feature based and example based summary statistics to demonstrate presence of concepts in the SAE representations, and motivate all four statistics well in Section 2.2. Unlike prior work on inferring CLIP concepts from SAEs, which tend to rely on singular feature space statistics, this makes the author's results a bit more comprehensive.
* The results from replacing CLIP features with SAE features and its impact on accuracy shows that the SAE conceptual features capture class discriminative information quite well. I think this result opens up more avenues on mechanistic interpretability research, as further exploration can be done into what makes a concept important vis-a-vis a class, what are the training dynamics of the concept learning for class discriminative performance (are there 'easier' concepts that are learned first e.g. winter before harder concepts e.g. christmas), and several other open research questions.

**Weaknesses:**

* The organization of the paper makes it hard for someone who is not very familiar with the field of sparse autoencoder based mechanistic interpretability of neural networks to follow the flow of ideas. For example Section 2 details the training procedure and the feature statistics calculated, but the experimental details are ommitted and then presented in Section 4. Similarly, the Intro and Section 2.1 cover a brief overview of SAEs and their use in intepretability research, but the actual related work section is not presented until Section 5. While I understand that the authors are eager to share their key results in an earlier section (especially since this is a intepretability focussed work), this organization made it more work for me as a reader to understand the paper.
* The scope of experiments performed and the subsequential observations is somewhat limited. The first experiment, showing presence of concepts in CLIP visual features, was already shown in principle by Fry et al and Daujotas et al. The result about CLIP attention maps focussing on features at different scales and regions is not surprising, but another sanity check for the SAE interpretability apporach. The second emperimental design is inspired from previous SAE results in LLMs e.g. Templeton et al, and extrapolated from a language (only) transformer setting to a multimodal setting.

**Questions:**

* "Specifically, we suggest taking all tokens including class (CLS) and image tokens from the second last attention block output and feeding them to SAE rather than limiting it to CLS token" Can the authors explain why they chose this particular layer for their analysis? Prior work has shown that self-supervised ViTs learn different invariances (hence, concepts) at different depths [1, 2], so one can expect that the SAE results from them would be very different. Are the results derived by the authors on this particular layer generalizable to more layers at different depoths, or did the authors chose this as a representative case study? If so, it would be interesting to see some ablations on how the results change with layer depth.
* I would be interested in seeing the results of experiment 3 (adaptation methods and SAE) with fine-tuning. Have the authors considered doing an experiment with fine-tuned CLIP for comparison? What are the potential bottlenecks in running such an analysis. And what results can be expected?
* Typo L146-147: Then we compute *how* frequently these features are activated for the given input.


# References

1. Walmer, M., Suri, S., Gupta, K., & Shrivastava, A. (2023). Teaching matters: Investigating the role of supervision in vision transformers. In Proceedings of the IEEE/CVF Conference on Computer Vision and Pattern Recognition (pp. 7486-7496).
2. Shekhar, S., Bordes, F., Vincent, P., & Morcos, A. (2023). Objectives matter: Understanding the impact of self-supervised objectives on vision transformer representations. arXiv preprint arXiv:2304.13089.

---

> ### Author Response · Authors · 2024-11-21
> **Response to Reviewer GMQj**
>
> *We appreciate the insightful comments and positive support with constructive feedback. We hope our responses address the reviewer's concerns.*
>
> ### **GMQj-W1**
> > The organization of the paper makes it hard to follow the flow of ideas.
>
> We deeply appreciate for providing detailed feedback regarding the structure. **We revised the overall structure** of the paper to make it easy to follow. We put related work before the main parts ($\S$ 2) and add clear explanations for experiment setups as a separate subsection before key findings in each section. Please refer to **GR1**.
>
> ### **GMQj-W2**
> > The scope of experiments and observations is somewhat limited.
>
> We clarify our contributions in the paper. Please see **GR3**.
> Key differences from existing work are:
> 1. Fry et al. showed the presence of interpretable SAE latents in CLIP and Daujotas et al. demonstrated the impact of SAE latents in controllable text-to-image generation. We propose to use **image tokens** for SAE latent analysis which allows **spatially localized understanding** of SAE latents and is easily transformable to a higher (image- / class- / dataset-) level of analysis. Fig. 4 and Fig. 9 highlight the effectiveness of localizing SAE latents.
> 2. Replacing the model output with SAE reconstructed one is inspired by previous works. We adopt this method to analyze the relationship between **interpretable concepts** and **downstream task-solving ability**, which addresses an important problem--understanding the **adaptation mechanisms** of foundation models.
>
> ### **GMQj-Q1**
> > Can the authors explain why they chose this particular layer for their analysis?
>
> We chose layer 11 as a representative case which we assume to contain rich information covering overall regions. **We added experiments for SAEs trained on different layers.** Please see **GR2.2**.
>
> As Fig. 9 shows, we find SAE latents show different patterns in different layers. The lower layer shows local attention, while the deeper layer shows global attention results. We believe the deeper layer is a reasonable choice for analyzing its influence to classification predictions. Moreover, to minimize the reconstruction error caused by replacing the model output with SAE reconstructed one in top-k masking experiments, we choose the second last layer instead of the very last layer. We identify more analysis on different layers as a promising direction for future work.
>
> ### **GMQj-Q2**
> > I would be interested in seeing the results of experiment 3 (adaptation methods and SAE) with fine-tuning.
>
> We chose prompt-based adaptation methods for controlled comparisons of zero-shot and adapted methods in the shared SAE latent space. After confirming the transferability of SAE in both methods (**GR2.4**), we share the SAE.
>
> To expand the experiments for fine-tuning method that directly updates the model parameters, we need to validate the transferability of the SAE. If SAEs are not sharable, we can either train separate SAEs and come up with methods for matching two latent spaces or jointly train sharable SAEs on two models for the top-k masking experiments conducted in our work. We consider examining the differences between different types of adaptation as a promising future research direction.
>
> ### **GMQj-Q3**
> >  Typo
>
> Thank you for pointing out the details. We revised the sentence in the manuscript.

---

> > ### Author Response · Authors · 2024-11-24
> >
> > Dear reviewer, thanks again for taking the time to reassess our work. Since the final days of the discussion phase are approaching, we wanted to reach out and ask if there are any further questions from your end. We hope that you like the re-organization and improvements of the manuscript, and the substantial additions to the experiments (both in the main paper and the appendix).

---

> > ### Comment · Reviewer_GMQj · 2024-11-25
> > **Response to author's rebuttal**
> >
> > Thank you for taking the time to provide a detailed rebuttal, as well as for adding new results and revising the paper. I think the new re-organization of the paper works significantly better than the original submission. I have consquently raised my rating to a clear accept.

---

> ### Author Response · Authors · 2024-11-30
> **Thank you!**
>
> We sincerely appreciate your positive assessment of our revised manuscript. If you have any additional questions or require further clarification, please let us know.

---

### Author Response · Authors · 2024-11-21
**Summary of rebuttal**

*We sincerely thank all four reviewers for their constructive feedback and valuable comments.*

Through **extensive revisions** to improve readability, the manuscript now offers enhanced clarity and structure. We would be deeply grateful if the reviewers could kindly take a fresh perspective on these changes. To allow for better exploration of our model, we now provide an **[interactive demonstration](https://huggingface.co/spaces/iclranonym/paper14240)** via Huggingface spaces.

**The strengths of our work, as highlighted by the reviews, include:**
* This work addresses an important but relatively under-explored problem and opens several research questions: understanding the inner workings of vision foundation models and adaptation mechanisms. (**GMQj**, **qx77**, **h3c2**)
* This work adopts SAE, a mechanistic interpretability tool that is widely studied for LLMs, to understand visual feature representations. (**67Ef**, **qx77**, **h3c2**)
* The analysis result demonstrates that the SAE model is successfully extracting interpretable concepts both in the patch- and image-level which contains class discriminative information. (**GMQj**, **67Ef**)

**Our responses to the reviews are as follows (4 parts):**
1. Paper re-organization and addressing ambiguities
2. Justification for SAE's reliability, design choices, and transferability
3. Clarification of contributions
4. Interactive demo: Highlighting key results

We provide brief summary of 4 parts of the responses in [**General Responses (GR)**](https://openreview.net/forum?id=imT03YXlG2&noteId=bE1RByHTPP) with references to the paper and the review items. We annotated the review items to the relevant paragraph of the revised paper as well. The reviewers can search for `{reviewer id}-{item id}` in the paper.

Once again, we appreciate the reviewers' considerate feedback and believe that our responses address their concerns effectively. Thank you for considering our rebuttal.

---

### Author Response · Authors · 2024-11-21
**General Responses to all reviewers**

**Please see related section in the revised paper.**

## **GR1. Paper re-organization and addressing ambiguities**

Our primary focus on the manuscript revision was on reconstructing the existing content to more clearly and effectively convey our findings and emphasize contributions. We added several figures and results to address the reviewers' concerns. Since we made extensive revisions, we provide a summary of the modifications.

* Terminologies
    * We named our SAE **PatchSAE**, emphasizing its strength in spatial attribution.
    * We use "SAE latents (latent directions)" to refer to the candidate concepts and "SAE latent activations" for the SAE ReLU activation output. ($\S$ 3.2)
* Main paper
    * $\S$ 1 Introduction, $\S$ 2 Related work, $\S$ 3 Training and evaluating SAE, $\S$ 4 Analyzing CLIP behavior via SAE, $\S$ 5 Discussion, and $\S$ 6 Conclusion
    * > GMQj-W1, qx77-W2, h3c2-W1, h3c2-Q1, h3c2-Q2
* Appendix
    * $\S$A SAE training details and ablation studies, $\S$B Additional results (including interactive web app) and supporting figures
    * > h3c2-W1

* Updated Figures (Main)
    * Fig.1 (overview): Clarified main contributions.
    * Fig.2 (SAE analysis method), Fig. 4 (segmentation mask): Improved readability
    * Fig.3 (evaluating SAE): Moved statistics scatter plot from the appendix, and added reference images.
        * > qx77-W1
    * Fig. 5 (top-k masking): Improved readability and moved full results in 11 datasets from the appendix.
    * Fig. 6 (**new**; analyzing adaptation): Elaborate "understanding adaptation methods via SAE" part
        * > 67Ef-W3, h3c2-Q1
    * Fig. 7 (**new**; scatter plot regions): Explain the scatter plot view of top-k class level activation comparisons

* Updated Figures (Appendix)

    * Fig. 9 (**new**; SAEs on different layers)
    * Fig. 10 (SAEs on different datasets): Moved from the main paper and revised.
    * Fig. 11 (**new**; SAE's transferability)
    * Fig. 12 (**new**; Screenshot for demo)
    * Fig. 13 (Multimodal latents): Added more cases.
    * Fig. 17 (new; Supplements new Fig. 7)
    * Fig. 22 (new; Supplements new Fig. 6)

## **GR2. Justification for SAE's reliability**

### **GR2.1.** Evaluating SAE ($\S$ 3.2, $\S$ 3.3, $\S$ 4.2, $\S$ A.1, and interactive demo)
* We explain how we validate SAE's reliability. We first evaluate reconstruction and sparsity metrics to that demonstrate our SAE follows the training objectives ($\S$ A.1). Then we use summary statistics and reference images with segmentation masks for qualitative analysis of SAE's validity as an interpretability tool ($\S$ 3.3). We provide an **[interactive demonstration](https://huggingface.co/spaces/iclranonym/paper14240)** to share more qualitative results. For classifications, we show that using our SAE (using all latents) successfully recovers the original classification performance ($\S$ 4.2).
* > 67Ef-W1, qx77-W1, qx77-Q1
### **GR2.2.** SAE training configurations ($\S$ A.1 and Table 2)
* We explain how we set training configurations with quantitative results from ablation studies.
* > 67Ef-W1, qx77-W3

### **GR2.3.** SAEs on different layers ($\S$ A.2 and Fig. 9)
* We train the SAEs on layers [2, 5, 8, 11(default)] and show comparisons. We find that the SAE latents on the deeper layers (i.e., closer to the output) provide semantically richer information with high confidence (high activation value) than the ones on the shallower layers.
* > GMQj-Q1, qx77-W3

### **GR2.4.** SAE's transferability on adapted models ($\S$ 4.1.1, $\S$ A.4, and Fig. 11)
* We intentionally choose MaPLe as an adaptation method to investigate the changes made by adaptation in the shared SAE latent space leveraging that MaPLe does not update model parameters. We demonstrate the transferability by comparing (backbone for SAE training, classification backbone) as (CLIP_SAE, CLIP), (MaPLe_SAE, CLIP), (CLIP_SAE, MaPLe), (MaPLe_SAE, MaPLe).
* > GMQj-Q2, 67Ef-W2, qx77-Q2

## **GR3. Clarification of contribution** ($\S$ 1, $\S$ 2, and $\S$ 5)

We discuss the scope of this work with broader implications and future work in $\S$ 5. We also clarify the main components of this paper in $\S$ 1 and state distinctive differences from existing work in $\S$ 2. In short, we introduce SAE to vision models and provide visual attribution of concepts to advance interpretability. We use our SAE to understand vision models' behavior on classification tasks and explain how adaptation affects the models' behavior in terms of interpretable concepts (latents).
* > GMQj-W2, 67Ef-W4, h3c2-W2


## **GR4. Interactive Demo**

We provide an **[interactive demonstration](https://huggingface.co/spaces/iclranonym/paper14240)** on Huggingface spaces with an anonymized account and quick instruction in Fig. 12. Users can select images, patches, models, and SAE latents. Based on the selection, the demo shows SAE latent activations, segmentation masks, and reference images.

---

### Author Response · Authors · 2024-12-04
**End of Rebuttal Phase**

Dear reviewers,
Dear AC,

we want to thank all reviewers again for the constructive feedback. Based on this feedback, we made a revision to the paper to enhance clarity and provide additional experimental evidence during the discussion phase **which was acknowledged by all reviewers and unanimously positively evaluated**.

**Thank you again for the time invested into reviewing our work.**

---

> ### Public Comment · ~Zeyu_Shi1 · 2025-03-31
>
> We thank reviewer LbiH for the valuable time and constructive feedback. Here are our responses to weaknesses and comments.
>
> **Q1：Lack of description of the optimization of t0 which is the most significant part of the framework.**
>
> **A1**: Thank you for the question. We have now provided a more detailed description of the threshold optimization process:
>
> As the knowledge distribution evolves during training, the threshold $t$ needs to be adjusted (Lines 310-315). Specifically, given a calibration set $S_{c}=\\{(x_i,y_i)\\}_{i=1}^N$ where $x_i$ represents the input and $y_i$ is the label, we perform the following steps at regular intervals (e.g., after each epoch):
>
> 1. The model performs inference on $S_c$ to obtain tuples $(c_i, n_i)$ for each sample, where $c_i$ indicates whether the model correctly predicts sample $i$, and $n_i$ is the negative log-likelihood.
>
> 2. Then we employ grid search to calculate threshold $t^*$: a set of thresholds $ \mathcal{T} =\\{t _ j\\} _ {j=1}^{M}$ is uniformly sampled from $[min(\\{n _ i\\} _ {i=1}^N),max(\\{n _ i\\} _ {i=1}^N)]$. The optimal threshold for the next round is selected by maximizing:
>
> $$
> t^*=\arg\max_{t\in\mathcal{T}}(TPR_t+TNR_t)
> $$
> where $TPR_t$ and $TNR_t$ denote the true positive and true negative rates, respectively, in classifying predictions as correct or incorrect based on the threshold $t$, evaluated on the calibration set $S_c$. The size of calibration set $S_c$ is detailed in Appendix A.
>
> We originally planned to include the detailed optimization for the threshold $t$ in the appendix. However, for the sake of clarity, we will instead incorporate these implementation details into the main text in Section 4.1.
>
> ------
>
> **Suggestion 1：Providing more detailed explanation of the t0 optimization process, including mathematical formulation, algorithmic steps, and practical guidelines for implementation across different models and tasks.**
>
> **R1：** The detailed explanation of the $t$ optimization process is provided in A1.
>
> ---
>
> **Sugge

---

### Meta-Review · Area_Chair_jmf6 · 2024-12-20

**Metareview:**

Post rebuttal, all of reviewers vote for acceptance. The AC checked all the materials and concurs that the paper has done a valuable exploration of using patch-wise sparse auto-encoders to interpret the CLIP visual representations, especially through the adaptation process. The paper received concerns about clarity and organization, but has been significantly improved during the rebuttal period, with authors paying great efforts providing new results and re-writing the draft. With these major revisions, the paper can be accepted. Please incorporate necessary changes in the final version.

**Additional Comments On Reviewer Discussion:**

All reviewers appreciate the authors' efforts during the rebuttal period, providing new results and re-organizing the paper for better clarity. As a result, all of them have raised the score and the paper receives a consensus for acceptance post-rebuttal.

---

### Decision · Program_Chairs · 2025-01-22

Accept (Poster)